# Enhanced multi-horizon photovoltaic power forecasting: A novel approach integrating ICEEMDAN decomposition with hierarchical frequency neural networks

Yaopeng Han[1,2,3,4], Chenxi Li[1], Siqi Chen[1], Jinghao Zhao[2,3,4], Yajun Tian[2,3,4], Jun Wang[1]*

**1** College of Computer Science and Technology, Shenyang University of Chemical Technology, Shenyang, Liaoning, China, **2** Extended Energy Big Data and Strategy Research Center, Qingdao Institute of Bioenergy and Bioprocess Technology, Chinese Academy of Sciences, Qingdao, Shandong, China, **3** Shandong Energy Institute, Qingdao, Shandong, China, **4** Qingdao New Energy Shandong Laboratory, Qingdao, Shandong, China

* wj_software@hotmail.com

## Abstract

As a crucial renewable energy source, solar PV power generation drives environmental protection and energy transformation. However, existing forecasting models struggle to accurately capture the complex dynamics of photovoltaic (PV) power, primarily due to monolithic modeling paradigms and inadequate representation of temporal information. To address these challenges, this paper proposes a novel hybrid model that leverages data decomposition and frequency-stratified prediction. The model employs the advanced ICEEMDAN algorithm to address complex non-stationarity. Additionally, it introduces a frequency-stratified heterogeneous network for precise component-wise modeling and integrates Improved Relative Positional Encoding (IRPE) to accurately capture temporal dependencies. To comprehensively evaluate model performance, this study employs quantile regression to generate probabilistic prediction intervals, using the median output as the baseline for point predictions. The model's performance is validated through ablation experiments and comparisons of single-step and multi-step predictions with recent benchmark models. The results indicate that the model excels under the MIMO strategy, achieving normalized nMAE values of 0.1142 and 0.1490 for 120-minute and 2880-minute forecasts on the DKASC and Solar I datasets, respectively, surpassing recent baseline models by 14.6% and 8.1%. Furthermore, the model's statistical stability and robustness are confirmed through 30 independent Wilcoxon signed-rank tests, as well as an uncertainty analysis conducted under various weather conditions. In summary, the model's high accuracy and stability provide robust support for power plant operations and planning.

**Data availability statement:** This study was supported by the Liaoning Provincial Department of Science and Technology in the form of a grant awarded to JW (Grant Number: 2025JH2/101300016) and the Liaoning Provincial Department of Education in the form of a grant awarded to JW (Grant Number: LJ212510149024). The specific roles of this author are articulated in the 'author contributions' section. The funders had no role in study design, data collection and analysis, decision to publish, or preparation of the manuscript.

**Funding:** This work was supported by the 2025 Liaoning Provincial Department of Science and Technology Applied Basic Research Program, under the project titled "Research on Intrusion Detection Technology and Intelligent Defense Strategies for Industrial Internet" (Grant Number: 1746669597594). This work was also supported by the 2025 Liaoning Provincial Department of Education Fundamental Research Project, focused on "Optimization Technologies for Key Metrics such as Energy Consumption and Coverage in Wireless Sensor Networks (WSN)." (Grant Number: LJ212510149024).

# 1 Introduction

Energy is the fundamental driving force behind modern global progress, and electricity demand is an important aspect of it [1]. Renewable energy is becoming increasingly attractive due to the high costs, limited reserves, and environmental impact of traditional energy sources. In recent years, the economic and environmental advantages of photovoltaic (PV) power generation have attracted widespread attention, and its market penetration rate has also significantly increased [2]. This has positioned PV to play a crucial role in meeting global energy demand and achieving dual-carbon goals. Despite the enormous potential of photovoltaic power generation, accurate estimation of its power generation remains a challenge due to the inherent instability and periodicity of the technology, which may affect the reliability and resilience of the power backbone network [3]. Therefore, it is imperative to improve the accuracy of photovoltaic power generation prediction.

In recent years, researchers have investigated a variety of technical approaches to achieve accurate predictions, encompassing physical methods, statistical techniques, and artificial intelligence [4]. While deep learning models, especially hybrid models that integrate signal decomposition techniques, have gained popularity and enhanced prediction accuracy to some degree [5], a thorough analysis of existing literature indicates that current research continues to face two fundamental limitations (a comprehensive discussion of related work is provided in Sect 2).

1. The current paradigm for signal decomposition and modeling has significant limitations in accurately capturing the complex dynamic characteristics of PV power. While signal decomposition is widely utilized, research often adheres to a homogenized modeling approach, applying a uniform model to both high-frequency components—characterized by peaks, valleys, and transient changes—and low-frequency components, which represent the fundamental trend. This method neglects the unique physical properties of each component, resulting in a sluggish model response to sharp fluctuations in PV power and ultimately limiting overall prediction accuracy [6]. Additionally, the choice of decomposition algorithm is critical for preserving these dynamic details. Compared to the commonly used CEEMDAN and VMD algorithms, the more advanced ICEEMDAN algorithm better suppresses modal aliasing and retains dynamic signal information [7].

2. In time series forecasting, the sequential information of data is critically important. The loss of positional information within sequences and inadequate model representation are significant challenges in this field. Although models based on the Attention Mechanism perform well in time series tasks, their standard attention modules cannot effectively capture the absolute or relative positional information of the input [8]. However, most existing studies either overlook this issue or utilize absolute positional encodings that can be rendered ineffective by linear transformations [9]. This limitation hinders the model's ability to learn temporal dependencies, ultimately constraining its predictive accuracy.

To address these limitations, this paper proposes a hierarchical frequency modeling network, as shown in Fig 1. In Step 1, the PCC [10] method is used to select climatic factors that significantly impact PV power. In Step 2, the ICEEMDAN algorithm decomposes the PV power sequence, generating a high-frequency error sequence and a low-frequency sum sequence. These sequences are combined with the selected climatic factors to form a PV power prediction dataset. In Step 3, the Conv1dBiGRU-IRPE-A model and LSTM model predict the high-frequency error sequence and low-frequency sum sequence, respectively. Finally, the model parameters are optimized using the PSO [11] algorithm, and the results of each frequency sequence are summed to obtain the final PV power prediction. To quantify the prediction uncertainty of the model, this study first employs quantile regression to generate probabilistic prediction intervals. Concurrently, the median of the model's output is used as the point forecast. Subsequently, the performance of the proposed ICIAL model was comprehensively evaluated on the DKASC and Solar I datasets. The evaluation included single-step forecasting, ablation studies, multi-step forecasting under multiple strategies (MIMO, Direct, and Recursive) [12], and a comparative analysis against recent baseline models. The results demonstrate that the ICIAL model achieves optimal performance under the MIMO strategy. Notably, in the multi-step MIMO forecasting experiments, the ICIAL model exhibited robust performance. It achieved a 10.2% reduction in nMAE compared to the ITransformer model [13] for the T+24 (120 min) task on the DKASC dataset, and a 5.9% reduction in nRMSE compared to the DA-GRU model [14] for the T+192 (2880 min) task on the Solar I dataset. To ensure the stability of the MIMO predictions, 30 rounds of Wilcoxon Signed-Rank Tests [15] were conducted. Specifically, 30 rounds of model training and prediction were performed on both the DKASC and Solar I datasets. The results reveal median p-values well above 0.05, with the proportion of significant differences being extremely low, peaking at 6.67%. This indicates high statistical consistency across multiple runs, confirming the greater performance and robustness of the ICIAL model. Furthermore, an uncertainty analysis conducted under various weather conditions shows the prediction intervals are both reliable, as measured by the Prediction Interval Coverage Probability (PICP), and sharp, as indicated by a narrow Prediction Interval Average Width (PIAW).

The contributions presented in this paper are as follows:

1. This study proposes a novel hybrid forecasting paradigm that combines signal decomposition with a frequency-stratified network. The proposed model, ICIAL, first employs ICEEMDAN to capture the complex fluctuation patterns of the PV power series. Subsequently, it utilizes a heterogeneous neural network integrated with Improved Relative Positional Encoding (IRPE) to perform stratified forecasting on the different frequency components, thereby significantly enhancing the model's ability to represent temporal features.
2. This study effectively addresses the limitations of traditional models in accurately capturing the complex dynamic characteristics of PV power. By applying targeted modeling techniques to both high- and low-frequency signals, the new paradigm significantly enhances the model's precision in capturing sharp fluctuations, including peaks, valleys, and transient changes.
3. This study systematically evaluated the model's performance and robustness across various multi-step forecasting strategies. Comparative experiments conducted in multiple scenarios confirmed the long-term forecasting performance advantages of the proposed model, particularly under the MIMO strategy. Furthermore, the Wilcoxon signed-rank test was employed to ensure the statistical consistency and reliability of the prediction results.
4. By introducing a quantile regression-based framework for uncertainty analysis conducted under various weather conditions, the model generates prediction intervals that demonstrate both high coverage probability (PICP) and robust sharpness (PIAW). This approach provides more comprehensive informational support for grid risk management and decision-making.

The remainder of this paper is organized as follows: Sect 2 introduces related works; Sect 3 details the core methodologies of the study, including the proposed ICIAL model and the validation techniques; Sect 4 provides a comprehensive description of the sources of the experimental data, the pre-processing procedures, and the methodologies employed for

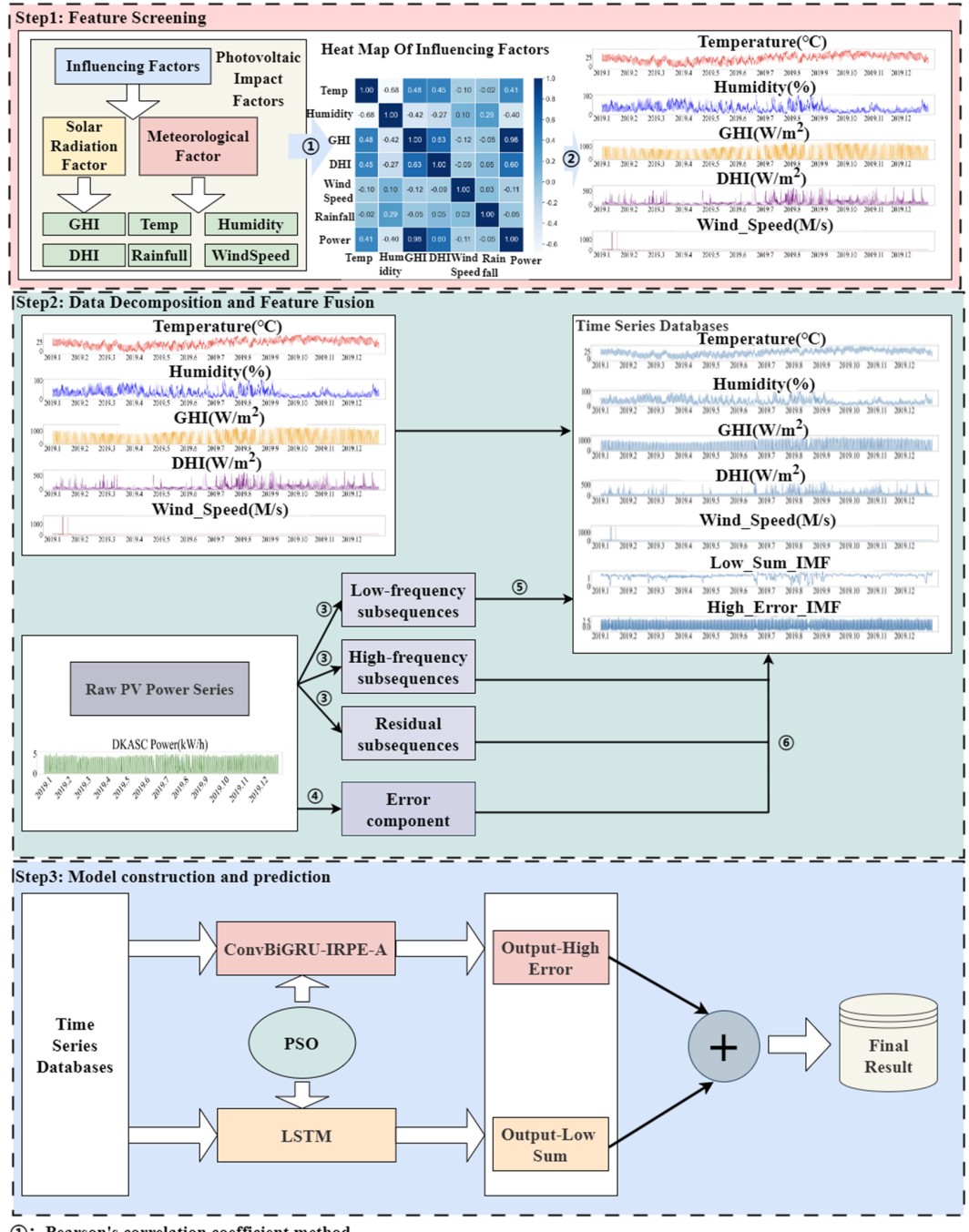

**Fig 1**. Prediction flowchart of the ICIAL model.

model training; Sect 5 presents a comprehensive experimental evaluation of the proposed ICIAL model, which includes performance comparisons against baseline models, stability validation via the Wilcoxon signed-rank test, and an assessment of its uncertainty quantification capabilities under various weather scenarios. Finally, Sect 6 summarizes the study and provides insights into potential directions for future research.

## 2 Related work

This section will provide a comprehensive review of prediction methods pertinent to this study, emphasizing the analysis of their strategies and limitations concerning hybrid model architectures, signal processing, and time series information modeling.

### 2.1 Evolution of mainstream PV prediction methods

The development of photovoltaic prediction technology has progressed from traditional methods to intelligent algorithms. Initially, physical and statistical methods were the predominant techniques. The application of physical methods measured physical data, Ding et al. [16] predict the PV output power based on meteorological data obtained from weather forecasts. However, physical models are highly dependent on data availability, which often limits their forecasting performance [17]. In contrast, statistical approaches offer enhanced flexibility and more rapid computational speeds. Vishal et al. [18] proposed a SARIMA-RVFL hybrid statistical model for very short-term solar PV power generation forecast. However, it turned out that physical and statistical methods cannot produce more accurate forecast results.

Recently, machine learning has shown great potential in PV power prediction, especially in the nonlinear correlation between PV power data and meteorological data [19]. Scott et al. [20] analysed a variety of machine learning models that are widely used in PV power forecasting, including models such as SVM and random forests, to provide insight into the advantages of different models in PV power forecasting. However, individual machine learning models often struggle to effectively capture intricate data relationships. Therefore, to improve the representation of complex time series, deep learning—particularly hybrid models that integrate various advantageous components—has increasingly become the predominant tool in the field of photovoltaic prediction [21]. Ali et al. [22] proposed two hybrid models (CNN-LSTM and ConvLSTM) to predict the power generation of PV plants. Tovar et al. [23] put forward a short-term power load forecasting model combining CNN and LSTM for photovoltaic power prediction with real data from a site in Mexico. However, none of these studies effectively address the non-stationarity of PV power data.

To address this issue, researchers have introduced signal decomposition techniques [5]. Feng et al. [24] proposed a PV power prediction model based on CEEMDAN-LSTM to improve the accuracy of PV power prediction results. Tao et al. [25] proposed sequence decomposition using VMD, where the decomposed subsequences and preprocessed historical meteorological data are fed into several LSTMs integrating one-dimensional convolution and attention mechanisms, respectively. The prediction results corresponding to each subsequence are then summed to obtain the final prediction result. Despite the application of decomposition techniques, current research has generally failed to adopt differentiated modeling approaches for the decomposed subsequences with varying physical characteristics, instead commonly employing homogeneous modeling paradigms. Table 1 summarizes these related studies, compares them with the ICIAL model proposed in this paper, and highlights the key limitations of existing methods.

### 2.2 Position representation in time series modeling

In photovoltaic time series forecasting, effectively representing the position of data points within the sequence to capture temporal dependencies is a core challenge, particularly in non-recurrent architectures based on the attention mechanism. As illustrated in Table 1, existing models address this issue in various ways. Traditional statistical models and recurrent neural networks implicitly model temporal order through their inherent sequential processing structures. However, to harness the powerful parallel computation and feature interaction capabilities of the attention mechanism, Transformer-based

**Table 1**. Comparative analysis of ICIAL model and advanced hybrid forecasting models.

| Ref. | Year | Model | Model Paradigm | Sig. Decomp. | Freq. Strat. | Pos. Enc. | Description |
|------|------|-------|----------------|--------------|--------------|-----------|-------------|
| [27] | 2025 | SARIMA | Stat | × | × | N/A | Cassica statistica mode; unabe to capture compex noninear reationships. |
| [28] | 2025 | Prophet-LSTM | Arch.Hybr | ✓ | × | Impicit | Seasonaity-focused hybrid; no advanced decomposition. |
| [29] | 2025 | CNN-LSTM | Arch.Hybr | × | × | Impicit | Captures noninearity; strugges with strong non-stationarity. |
| [30] | 2025 | BiLSTM-Attention | Arch.Hybr | × | × | Impicit | Enhances tempora feature extraction for soar forecasting; sensitive to non-stationarity. |
| [31] | 2025 | Waveet-BiLSTM | Deco.Hybr | ✓ | × | Impicit | Waveet decomposition and dua-attention BiLSTM; weaker adaptabiity than EMD-based methods. |
| [24] | 2023 | CEEMDAN-LSTM | Deco.Hybr | ✓ | × | Impicit | CEEMDAN-based LSTM ensembe; suffers from peak underestimation. |
| [32] | 2025 | VMD-TCN-GRU | Deco.Hybr | ✓ | × | Impicit | Combines VMD with TCN-GRU hybrid for therma prediction; appies uniform compex mode to a components. |
| [25] | 2024 | VMD-CNN-LSTM | Deco.Hybr | ✓ | × | Impicit | VMD-based CNN-LSTM ensembe; uniform processing of a subsequences. |
| [33] | 2025 | ET-EEMD-GRU | Deco.Hybr | ✓ | × | Impicit | Improved EEMD-GRU hybrid; a decomposed components modeed uniformy. |
| [34] | 2024 | N-BEATS Hybrid | Arch.Hybr | × | ✓ | N/A | Appies hybrid N-BEATS to PV forecasting; ess effective on strongy non-stationary signas. |
| [35] | 2025 | AT-Informer-AT | Singe.DL | × | × | Absoute | Improved Informer for PV forecasting; no signa decomposition. |
| [36] | 2025 | EMD-VMD-Autoformer | Deco.Hybr | ✓ | × | Absoute | Autoformer with secondary decomposition; strugges with reative tempora dynamics due to rigid absoute encoding. |
| [37] | 2025 | VMD-SSA-Transformer-LSTM | Deco.Hybr | ✓ | × | Absoute | Combines VMD with Transformer-LSTM predictor; positiona encoding remains a imitation. |
| [38] | 2024 | CEEMDAN-SE-CNN-GRU | Deco.Hybr | ✓ | × | Impicit | Groups CEEMDAN components by compexity (SE) for CNN-GRU prediction; reies on impicit positiona encoding, risking tempora information oss. |
| **This study** | 2025 | **ICIAL** | **Freq. Strat** | ✓ | ✓ | **IRPE** | **Nove framework combining ICEEMDAN, Heterogeneous Network, and IRPE to sove dua bottenecks of peak underestimation and tempora information oss.** |

Note: Stat=Statistica; Arch.Hybr=Architecture Hybrid; Deco.Hybr=Decomposition-Hybrid; Singe.DL=Singe Deep Learning; Sig. Decomp.=Signa Decomposition; Freq. Strat.=Frequency-Stratified; Pos. Enc.=Positiona Encoding.

models must implement explicit positional encoding schemes [8]. Currently, the most common approach is absolute positional encoding. However, a substantial body of research indicates that the precise positional information encoded in this manner may be weakened or lost after multiple layers of non-linear transformations, thereby affecting the model's ability to learn temporal dynamics [9].

In contrast, relative positional encoding (RPE), which directly models the "query-key" relative temporal relationships between data points, has proven to be a more robust and suitable strategy for capturing dynamic changes [26]. However, in the field of photovoltaic forecasting, this crucial difference has not been adequately addressed. As clearly illustrated in the comparison presented in Table 1, most existing advanced hybrid prediction models rely on implicit positional representations or basic absolute positional encoding. To address these limitations and further enhance the model's accuracy in capturing temporal dynamics, this paper introduces an improved relative positional encoding (IRPE) mechanism.

In summary, existing photovoltaic prediction models, as shown in Table 1, struggle to accurately capture the complex dynamics of power output due to dual bottlenecks in signal processing and temporal representation, which limits both prediction accuracy and reliability. To address these challenges, this research proposes a novel ICIAL prediction framework. This framework aims to resolve the issue of inaccurately capturing dynamic characteristics by introducing a frequency-hierarchical heterogeneous network architecture designed for the targeted modeling of signal components at different frequencies. Additionally, it incorporates an improved relative positional encoding (IRPE), which significantly enhances the model's capacity to represent dynamic temporal dependencies, ultimately achieving greater accuracy and robustness in multi-step prediction tasks.

## 3 Methods

This section introduces the methodology of the study. First, a heterogeneous prediction framework based on ICEEMDAN decomposition (ICIAL) is described, including its overall workflow and the function of each module. Finally, the Wilcoxon signed-rank test, which is employed to ensure the statistical significance of the results, along with the quantile regression method for quantifying prediction uncertainty, is presented.

### 3.1 ICEEMDAN algorithm

The ICEEMDAN algorithm refines EMD by adding adaptive white noise . This approach reduces mode mixing, ensures frequency continuity across scales, and improves both the accuracy and interpretability of the decomposition. In the iterative decomposition process of ICEEMDAN, each IMF is extracted using the current residual for subsequent steps, rather than by reconstructing the original dataset. This mechanism prevents the introduction of future data when decomposing individual time series. Additionally, to avoid incorporating future information while processing long sequences, this study employs a sliding window technique in conjunction with the ICEEMDAN algorithm, which further enhances prediction accuracy. The ICEEMDAN algorithmic process [39] is described in Table 2:
where $a^i$ and $r_1$ are shown in Eqs (1) and (2):

$$a^i = a + \alpha_0 \cdot \text{std}(a) \cdot w^i \tag{1}$$

$$r_1 = \frac{1}{L} \sum_{i=1}^{L} \text{mean}(a^i) \tag{2}$$

In both equations, $a^i$ represents the $i$-th ensemble member generated by adding the $i$-th set of white noise $\omega^i$ (which has a zero mean and unit variance) to the original sequence $a$. Here, $\alpha_0$ is the weighting constant for the initial noise level, $\text{std}(a)$ denotes the standard deviation of $a$, $r_1$ is the first residual following the initial decomposition stage of ICEEMDAN, and $L$ represents the ensemble size.

**Table 2**. ICEEMDAN algorithm steps with implementation details.

| Step | Description |
|---|---|
| 1 | Generate Group I ensembles $\{a^i\}_{i=1}^I$ (where $I = 50$) by adding zero-mean white Gaussian noise $w^i$ with a standard deviation of $\epsilon_0 \cdot std(a)$ (where $\epsilon_0 = 0.2$) to the original sequence $a$. These ensembles are denoted as $a^i = a + \epsilon_0 \cdot std(a) \cdot w^i$ for $i = 1, \dots, I$. |
| 2 | Obtain the first residual $r_1$ by averaging the local means of the ensembles $\{a^i\}_{i=1}^I$ obtained from a single EMD decomposition. Specifically, for each ensemble $a^i$, perform EMD to get the first intrinsic mode function (IMF) component $IMF_1(a^i)$, and then the local mean is approximated as $a^i - IMF_1(a^i)$. The first residual is $r_1 = \frac{1}{I}\sum_{i=1}^I(a^i - IMF_1(a^i))$. |
| 3 | Calculate the first IMF as $IMF_1 = a - r_1$. |
| 4 | For $k = 1, 2, \dots,$ maximum number of IMFs $- 1$: |
| 5 | Generate noise modes $E_k$ by performing EMD on white noise, then generate the $(k + 1)$-th stage ensembles $\{r_k^i\}_{i=1}^I$ by adding zero-mean white Gaussian noise with adaptive attenuation: $r_k^i = r_k + \epsilon_k \cdot std(a) \cdot E_k$ (where $\epsilon_k = \epsilon_0 \cdot (0.8^k)$) for $i = 1, \dots, I$. |
| 6 | Calculate the $(k + 1)$-th residual $r_{k+1}$ by averaging the local means of the ensembles $\{r_k^i\}_{i=1}^I$ obtained from a single EMD decomposition: $r_{k+1} = \frac{1}{I}\sum_{i=1}^I(r_k^i - IMF_1(r_k^i))$. |
| 7 | Compute the $(k + 1)$-th IMF as $IMF_{k+1} = r_k - r_{k+1}$. |
| 8 | Stop the decomposition when: (1) the residual $r_{k+1}$ has less than 3 extrema; (2) the variance of $r_{k+1}$ is less than 1e–10; or (3) the maximum number of IMFs is reached. |
| 9 | End. |

Note: Variables used are: $r_k$ (residuals), $E_k$ (noise modes), $\epsilon_k$ (adaptive noise attenuation), and $IMF$ (intrinsic mode function). The parameters were chosen based on empirical testing and common practices [39,40].

### 3.2 PSO algorithm

The Particle Swarm Optimization (PSO) algorithm is a group intelligence method that simulates the social foraging behavior of bird flocks. This approach is based on the observation that birds tend to fly in a swarm, with each bird following a set of simple rules to reach an optimal solution. In particular, the algorithm guides the movement of particles within the search space by modifying their velocities and positions in order to bring them as close as possible to the optimal solution. The formulae for calculating the velocity and position of the particulates are as follows [11].

$$v_y^{k+1} = w v_y^k + c_1 r_1(p\_best_y^k - x_y^k) + c_2 r_2(g\_best_z^k - x_z^k) \tag{3}$$

$$x_y^{k+1} = x_y^k + v_y^{k+1} \tag{4}$$

Where $v_y^k$ and $v_y^{k+1}$ are the velocity values of the particles in the neighbouring moments; $w$ is the inertia weight; $x_y^k$ and $x_y^{k+1}$ are the position values of the particles in the neighbouring moments; $c_1$ and $c_2$ are factors that contribute to the process of learning; $r_1$ and $r_2$ are the randomly generated numbers ranging from 0 to 1; $pbest_y^k$ and $gbest_z^k$ are the individual and global historical optimal position values at time $k$, respectively.

Fig 2 illustrates the detailed algorithmic flow of the PSO algorithm for the ICIAL model applications.

### 3.3 Wilcoxon signed-rank test method

This study employs the Wilcoxon signed-rank test [15] to ensure statistical consistency and mitigate the randomness inherent in deep learning models. This non-parametric statistical method is designed to assess whether significant differences exist between two paired datasets, making it particularly suitable for scenarios where data distributions do not conform to normality assumptions.

The core formulation of the Wilcoxon signed-rank test is as follows: Consider two paired datasets $X_i$ and $Y_i$ ($i = 1, 2, \dots, n$), with differences defined as $d_i = X_i - Y_i$. Samples with $d_i = 0$ are excluded, and the absolute differences $|d_i|$ are

Particle Swarm Optimisation Algorithm Flow

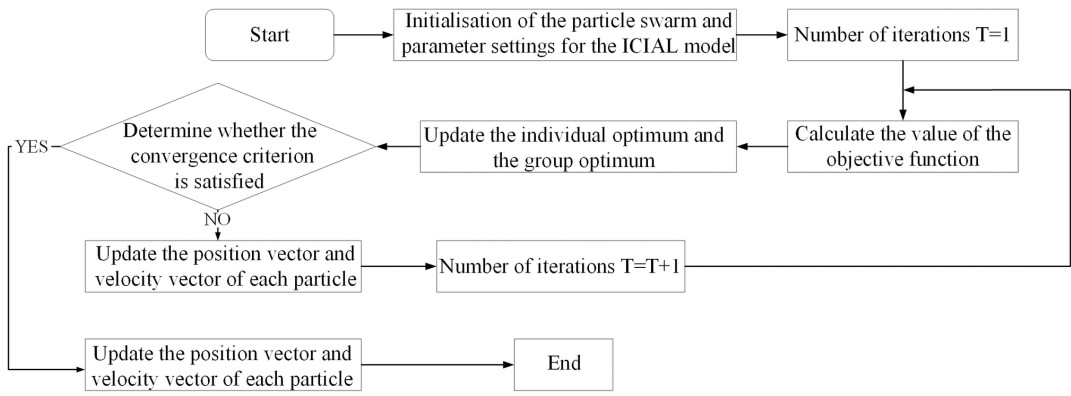

**Fig 2**. **Flow chart of particle swarm optimisation algorithm.**

ranked in ascending order, denoted as $R_i$. The positive rank sum $W^+$ and negative rank sum $W^-$ are calculated based on the signs of $d_i$:

$$W^+ = \sum_{d_i>0} R_i, \quad W^- = \sum_{d_i<0} R_i \tag{5}$$

The test statistic $W$ is taken as the smaller of $W^+$ and $W^-$. For large samples ($n > 20$), $W$ approximates a normal distribution, with its standardized form given by:

$$Z = \frac{W - \mu_W}{\sigma_W} \tag{6}$$

where $\mu_W = \frac{n(n+1)}{4}$ and $\sigma_W = \sqrt{\frac{n(n+1)(2n+1)}{24}}$.

The p-value is calculated based on the standardized statistic $Z$. For a two-tailed test, the p-value is determined using the cumulative distribution function (CDF) of the standard normal distribution:

$$p = 2 \times [1 - \Phi(|Z|)] \tag{7}$$

where $\Phi(\cdot)$ is the CDF of the standard normal distribution, and $p$ represents the probability of observing the test statistic under the null hypothesis. The significance level ($\alpha$), typically set at 0.05, is a predefined threshold used to compare with the p-value to assess statistical significance.

This study employs p-values and a significance level $\alpha$ as evaluation criteria. The p-value represents the probability of observing a test statistic under the null hypothesis, while $\alpha$ serves as the significance threshold. A p-value less than $\alpha$ indicates a significant difference, whereas a p-value greater than $\alpha$ suggests no significant difference and indicates statistical consistency. To enhance the robustness of evaluations, the median p-value is employed as a key metric. This measure reflects the central tendency of p-values across multiple experiments, is less sensitive to outliers, and more reliably represents the statistical consistency of model performance. This approach is particularly effective when analyzing repeated runs, ensuring that evaluations are not influenced by random fluctuations [41].

## 3.4 Uncertainty quantification and interval prediction method

To address the limitation of point prediction in quantifying the inherent uncertainties of PV power generation, this study introduces an uncertainty quantification framework based on quantile regression. This method is particularly suitable for PV forecasting because it eliminates the need to assume the error distribution— a key advantage given that prediction errors in PV systems typically exhibit non-Gaussian characteristics [42]. By generating probabilistic prediction intervals, the approach provides critical information for risk assessment and operational decision-making in power systems.

**3.4.1 Quantile regression and pinball loss function.** Quantile regression is a statistical technique used to estimate the conditional quantiles of a response variable [43]. It can model the entire conditional distribution, rather than just focusing on the conditional mean as in traditional regression. This offers significant advantages in photovoltaic power forecasting: First, it does not require assumptions about the distribution of prediction errors, thus demonstrating strong robustness when dealing with the skewed and heavy-tailed error distributions commonly found in photovoltaic data due to weather fluctuations. Second, by modeling the upper and lower bounds at a given confidence level, it can directly generate prediction intervals. To train a model for quantile prediction, a quantile loss function (also known as the pinball loss function) must be used. This function applies asymmetric penalties for overestimation and underestimation, guiding the model to learn specific conditional quantiles. For a target quantile $q$ (where $q$ is between 0 and 1), the loss function $L_q$ is defined as follows:

$$L_q(y, \hat{y}) = \begin{cases} q \cdot (y - \hat{y}) & y \geq \hat{y}, \\ (1 - q) \cdot (\hat{y} - y) & y < \hat{y} \end{cases} \tag{8}$$

where $y$ denotes the actual value, $\hat{y}$ represents the predicted quantile value, and $q$ is the target quantile.

**3.4.2 Interval prediction evaluation metrics.** To quantify the performance of the generated prediction intervals, two standard metrics are employed: Prediction Interval Coverage Probability (PICP) and Prediction Interval Average Width (PIAW).

PICP measures the percentage of actual values that fall within the prediction intervals, reflecting the reliability of the predictions. A higher PICP indicates more reliable intervals. The formula for PICP is:

$$\text{PICP} = \frac{1}{N} \sum_{i=1}^{N} I(y_i \in [L_i, U_i]) \times 100\% \tag{9}$$

where, $N$ is the total number of samples, $y_i$ is the $i$-th actual value, $[L_i, U_i]$ is the prediction interval for the $i$-th sample, $I(\cdot)$ is the indicator function.

PIAW measures the average width of the prediction intervals. Under the premise of a satisfactory PICP, narrower intervals imply more precise and sharper predictions. The formula for PIAW is:

$$\text{PIAW} = \frac{1}{N} \sum_{i=1}^{N} (U_i - L_i) \tag{10}$$

where $U_i$ and $L_i$ are the upper and lower bounds of the prediction interval for the $i$-th sample, respectively.

## 3.5 High frequency forecasting model

The high-frequency error sequence information displays significant fluctuations and high-frequency variation characteristics. Consequently, the hybrid model ConvBiGRU-IRPE-A captures these notable high-frequency fluctuations more effectively by integrating the one-dimensional convolutional layer, the BiGRU unit, the IRPE mechanism, and the attention mechanism, thereby enhancing the accuracy of the prediction results.

 

**3.5.1 Attentional mechanism.** The Attention Mechanism Model [8]is one of the most widely used attention models in current. It calculates the similarity by vector dot product and scales the result in accordance with the respective magnitude of the input vectors. Subsequently, Softmax function is applied for normalisation to generate the attention weights. Ultimately, the aforementioned weights are employed to weigh the value vectors, thereby forming the attention score matrix. The following is the standard formula for Attention process calculations:

$$\begin{cases} \text{Attention}(Q, K, V) = \text{Softmax}\left(\frac{QK^T}{\sqrt{d}}\right) V \\ Q = X \times W_q \\ K = X \times W_k \\ V = X \times W_v \end{cases} \tag{11}$$

Where $\text{Attention}(Q, K, V)$ is a self-attention score that is calculated in accordance with the specified parameters; $Q$ represents the variable used to formulate the query; The variable designated $K$ refers to the principal vector; $V$ represents the magnitude of the values contained within the value vector; $X$ indicates the value of the input vector. The matrices $W_q$, $W_k$, and $W_v$ correspond to trainable linear transformation matrices for the $Q$,$K$, and $V$ vectors,respectively. $d$ denotes the dimension of the input vector.

**3.5.2 Improved relative position encoding (IRPE).** Absolute position encoding is a method of generating a unique representation for each position in a sequence using the following formula:

$$T = (x_1 + p_1, \cdots, x_i + p_i) \tag{12}$$

Where: $T$ denotes the model input vector after position coding; $x_i$ denotes the data embedding vector of the $i$-th position; $p_i$ indicates the absolute position coding vector of the $i$-th position.

Using absolute position encoding, the formula for calculating the encoded value of the $i$-th position in relation to the $j$-th dimension in the model is as follows [9]:

$$S_{i,j} = x_i W_Q W_K{}^T x_j{}^T + p_i W_Q W_K{}^T p_j{}^T + x_i W_Q W_K{}^T p_j{}^T + p_i W_Q W_K{}^T x_j{}^T \tag{13}$$

Absolute positional coding primarily considers the absolute positions of elements within a sequence. In contrast, relative positional coding allows the model to effectively account for the positional relationships between elements during the learning process, particularly focusing on the relationships across different sequences. This approach resolves the limitations of analyzing elements solely within a single sequence.

This paper proposes an improved XLNet-style [44] position coding method that introduces two different trainable relative position identification parameters that vary linearly during vector computation. Following this approach, the parameters are adjusted in real-time throughout the training phase, aiming for continuous optimization via backpropagation within the network. In each training round, the model updates the trainable parameters $u$ and $v$ based on the training weights. This will result in the updating of parameter values and generation of new parameters $u_1$ and $v_1$. The newly generated parameter values will then be employed as initial parameter values for the subsequent training round. As the model training progresses, the weight values will continue to evolve, enabling the parameters $u$ and $v$ to be progressively refined until the model is optimally trained, reaching $u_n$ and $v_n$. Additionally, the positional information between the sequences is adapted to facilitate the dynamic adjustment of the $u$ and $v$ values. This process optimises relative position coding, thereby improving its ability to capture detailed relative position relationships between different time points. In this paper,

the improve relative position coding formula for linear transformation of trainable parameters is introduced as follows:

$$S_{i,j} = x_i W_Q W_K^T x_j^T + u_i W_K^T x_j^T + x_i W_Q W_K^T R_{i-j}^T + v_i W_K^T R_{i-j}^T \qquad (14)$$

Where: $S_{i,j}$ denotes the vector product of calculating the $i$-th element of $Q$ and the $j$-th element of $K$; $R_{i-j}$ denotes the relative positional distance as it relates to the query vector $q_i$ as well as the key vector $k_j$; $u_i$ and $v_i$ are two parameter vectors that can be trained to vary.

### 3.5.3 ConvBiGRU-IRPE-A.

The ConvBiGRU-IRPE-A model integrates several advanced components to achieve precise predictions. First, a Conv1D serves as the initial feature extractor, capturing local fluctuation patterns in photovoltaic power caused by abrupt weather changes. Subsequently, a BiGRU processes the sequence in both forward and reverse directions to capture comprehensive contextual information, thereby modeling long-term dependencies and overall trends [45]. Building on this, an attention mechanism dynamically assigns weights to different time steps, allowing the model to focus on the most critical historical information. Finally, to address the limitations of standard attention mechanisms in perceiving sequence order, this study designed an IRPE. By dynamically encoding the relationships between time steps, the IRPE significantly enhances the model's representation of temporal dynamics.

Fig 3 illustrates the workflow of the model. Initially, the high-frequency error sub-series is input into a Conv1D layer for local feature extraction, followed by a fully connected layer for feature integration. The processed features are then passed to a stacked BiGRU network to capture bidirectional temporal dependencies. A Dropout layer is inserted between consecutive BiGRU layers to randomly deactivate a fraction of the neurons, effectively reducing model complexity and mitigating overfitting. Subsequently, the IRPE mechanism injects dynamic relative position information into the output states of the BiGRU. These temporally informed states are sent to an attention layer that computes and assigns weights to emphasize critical time steps. Finally, the weighted features are mapped to the final output space via another fully connected layer, generating the predicted values for the high-frequency error series.

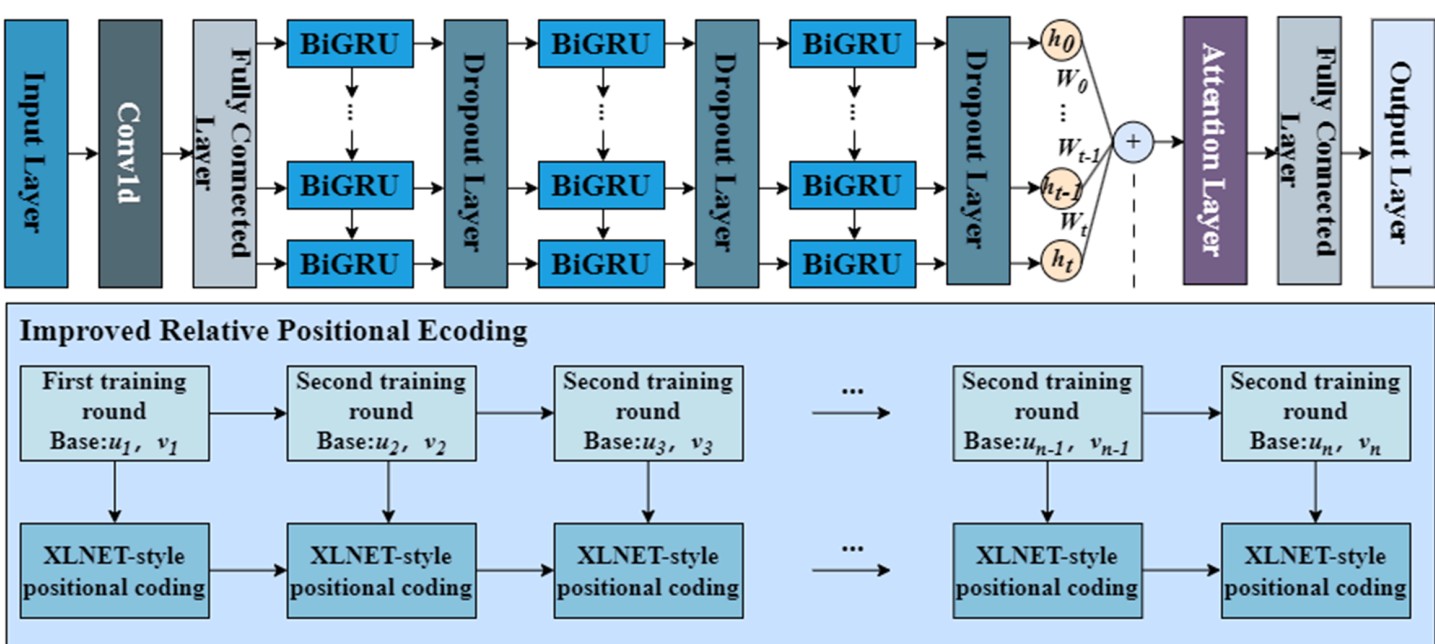

**Fig 3**. **The architectural design of the ConvBiGRU-IRPE-A model.**

Fig 3 is divided into two sections. The upper section illustrates the primary data flow of the model, detailing the core components from the Input Layer to the Output Layer, which include the Conv1D layer, the BiGRU network, Dropout layers, and the Attention Layer. In this diagram, $h_t$ represents the hidden state output of the BiGRU at time step $t$, while $W_t$ denotes the corresponding weight computed for this state by the attention mechanism. Meanwhile, the lower section explains the principles of the IRPE mechanism. This mechanism dynamically generates XLNET-style relative position encodings by continuously updating two trainable base parameters, $u$ and $v$, during each training epoch. This process significantly enhances the model's ability to capture temporal dynamics.

## 3.6 Low frequency forecasting model

Low-frequency summed series are typically characterised by slow data changes and significant trends or periodic changes. It can be reasonably deduced that the forecasting model for low-frequency summed series should adopt the traditional LSTM structure, as this will enable the LSTM model to more effectively capture and store the characteristics of these series [46].

**3.6.1 LSTM.** LSTM was first conceptualised by Hochreiter in collaboration with Schmi-dhuber. Its core idea is to alleviate the gradient vanishing problem by using the 'gate' structure with short-time memory and the unit state with long-time memory. The architectural of LSTM is illustrated in Fig 4, and the specific computational procedures of different gates are as follows [47]:

$$f_t = \sigma(W_f x_i + U_f h_{t-1} + b_f) \tag{15}$$

$$i_t = \sigma(W_i x_i + U_i h_{t-1} + b_i) \tag{16}$$

$$\widetilde{C}_t = \tanh(W_c x_t + U_c h_{t-1} + b_c) \tag{17}$$

$$C_t = f_t \circ C_{t-1} + i_t \circ \widetilde{C}_t \tag{18}$$

$$O_t = \sigma(W_o x_i + U_o h_{t-1} + b_o) \tag{19}$$

$$h_t = o_t \circ \tanh(C_t) \tag{20}$$

$h_{t-1}$ and $h_t$ are the states of the hidden layer that have preceded and succeeded one another in time; $C_{t-1}$ and $C_t$ are the previous and current unit memory information, respectively; $f_t, i_t, \widetilde{C}_t, O_t$ are the vectors that represent the degree of activation of respective components of the neural network, namely the forgetting gate, the input gate, the cell input, and the output gate; $W_K, U_K, b_k$ denote the weight matrix of the input data vector $x_i$, respectively; $h_t$ is the hidden-state vector; Bias $f, i, c, o$ representing the states of different gates or cells, where $k = f, i, c, o$ and $\circ$ denote the Hadamard product; $\sigma$ and $\tanh$ denote the sigmoid activation and hyperbolic tangent functions, respectively.

## 4 Experiment

Two distinct datasets were employed in this study, and pre-processing operations, including the filling of missing values, handling of outliers, and normalization of data, were conducted on each of these two datasets. Subsequently, the ICEEM-DAN algorithm was applied to the pre-processed datasets. The goal of these operations is to ensure data integrity and reliability for subsequent analysis.

## 4.1 DKASC dataset

The Desert Knowledge-Based Energy Research and Solar Technology Application Centre (DKASC) is situated in the arid desert environment of the Northern Territory's Alice Springs. Despite the region's lack of precipitation, it boasts one of the highest solar resources in Australia. The dataset is publicly available for consultation in the relevant literature [48]. The data selected for analysis in this study was collected over the period between the first of January, 2019, and the

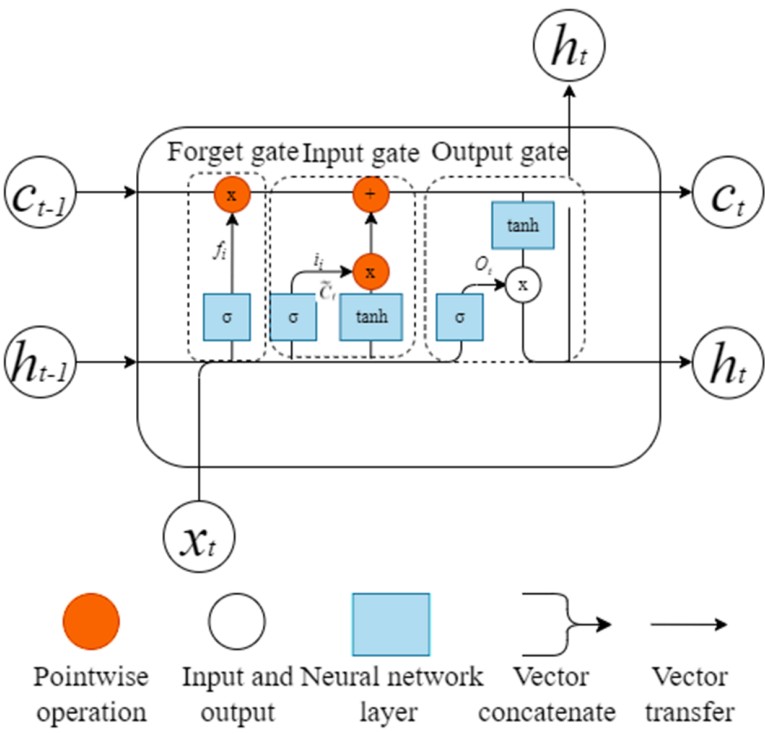

**Fig 4**. **Structure of the LSTM unit.**

thirty-first of December, 2019. Table 3 presents the supplementary technical particulars of the PV sites. The temporal resolution of the data is 5 minutes. The experimental data set was composed of 80% of the full-year 2019 data, the validation-set 15%, and the test-set 5%. The PV power prediction was trained using these datasets, and the resulting predictions were compared with the actual data.

The value of the PCC of the combined PV power and weather influences was calculated [10] and the heat map were generated. The results demonstrate a notable interdependence between PV power and irradiance.According to the heat map, the absolute value of the correlation coefficient of the influencing factors is greater than 0.1, the following were selected as the weather influencing factors for the experiment: global horizontal irradiance, scattered radiation, relative humidity, air temperature and wind speed.

**Table 3**. **Technical parameters of PV power generation system.**

| Technical Specification | Values |
|---|---|
| Array Rating | 5.2 kW |
| Panel Scoring | 260 Watts |
| Number of panels | 20 |
| Panel Type | Silicon Classic P260 |
| Inverter Size/Type | 250W, Enphase M250-72 Microinverter (20 units) |
| Mounting Completion | Thursday, 9th June 2016 |
| Array Tilt/Azimuth | Pitch = 20, Azimuth = 0 (Sun North) |

## 4.2 SolarI dataset

SolarI dataset was derived from the Renewable Energy Power Generation Forecasting Competition [49], which was organised by the State Grid of China. In comparison to the DKASC dataset, the SolarI dataset is characterised by a paucity of data and a prevalence of anomalous weather conditions. The dataset comprises PV data from eight solar stations located within a single geographical region of China, spanning the period between 2019 and 2020. The 2019 PV power generation data from the Solar Station 1 dataset, which has a nominal capacity of 50 MW, is employed as the experimental dataset, with a time step of 15 minutes for each data point. The dataset has been divided into three distinct subsets: the training-set, comprising 70% of the full-year 2019 data; the validation-set, comprising 10% of the full-year 2019 data; and the test-set, comprising the remaining 20% of the full-year 2019 data. The data are employed in order to train the photovoltaic power prediction model, and thus generated predictions are then compared with the actual data.

## 4.3 Preprocessing

The data was pre-processed using characteristic equations to fill in missing values and remove anomalies. This was done in order to avoid the inclusion of any missing or anomalous values in the data set [50]. Subsequently, the data underwent normalisation and transformation into a uniform standard form, thus facilitating subsequent modelling and experimentation. Then, the ICEEMDAN algorithm has been utilised for the purpose of decomposing PV power series within this standardised dataset. This has enhanced the precision and reliability of data analysis.

**4.3.1 Feature engineering.** It is unavoidable that the raw data will contain missing values as a consequence of the inevitable wear and tear caused by the operation and maintenance of equipment. In the event that the aforementioned missing values are situated within a span of 10 consecutive data points, they are duly filled through the utilisation of linear interpolation, employing the surrounding data as a reference. In cases where more than 10 consecutive data points are absent, for instance as a consequence of equipment malfunction or scheduled maintenance, the data for the days in question are removed. Following the aforementioned data pre-processing, the DKASC dataset comprises 104,256 data rows, while the SolarI dataset contains 35,040 data rows.

**4.3.2 Data normalization.** To eliminate the impact of different data scales and to account for the distinct physical boundaries of photovoltaic (PV) power data (e.g., zero power at night), this study employs Min-Max Scaling for data preprocessing [10]. This method linearly maps the original data, $x$, to the interval [0, 1] using the following formula:

$$x^* = \frac{x - \min(x)}{\max(x) - \min(x)} \tag{21}$$

Where $x$ represents the original data, $x^*$ is the normalized data, and $\min(x)$ and $\max(x)$ are the minimum and maximum values of the original dataset, respectively. This method linearly maps the data to the interval [0,1], preserving the shape of the original distribution. This approach is particularly well-suited for PV forecasting as it accounts for the distinct physical boundaries of the data (e.g., zero power at night), which is beneficial for the subsequent quantile regression interval forecasting task. All predicted results are inversely normalized prior to the calculation of evaluation metrics.

**4.3.3 ICEEMDAN decomposition results of datasets.** The PV power sequence was decomposed in a stepwise manner using the ICEEMDAN algorithm. In order to obtain the multi-scale modal components of the power series in its original form, the Signal-to-Noise Ratio was established at 0.2, with the maximum number of iterations designated as the value of 100. This resulted in the identification of 17 sets of IMFs and 1 set of Res. As shown in Fig 5, the decomposition of the 2019 PV power series for the DKASC dataset is presented, which comprises 104,256 data points. The zero-crossing rate [51] of the signal is used to determine the frequency of the mode decomposition sequence(i.e. the frequency with which the signal crosses the zeros in a specified time interval). Subsequently, the over-zero rate of each decomposition sequence is calculated in accordance with Eq (22).The sequences are divided into nine groups of high-frequency sequences and eight groups of low-frequency sequences, as well as a Res sequence that cannot be further decomposed,

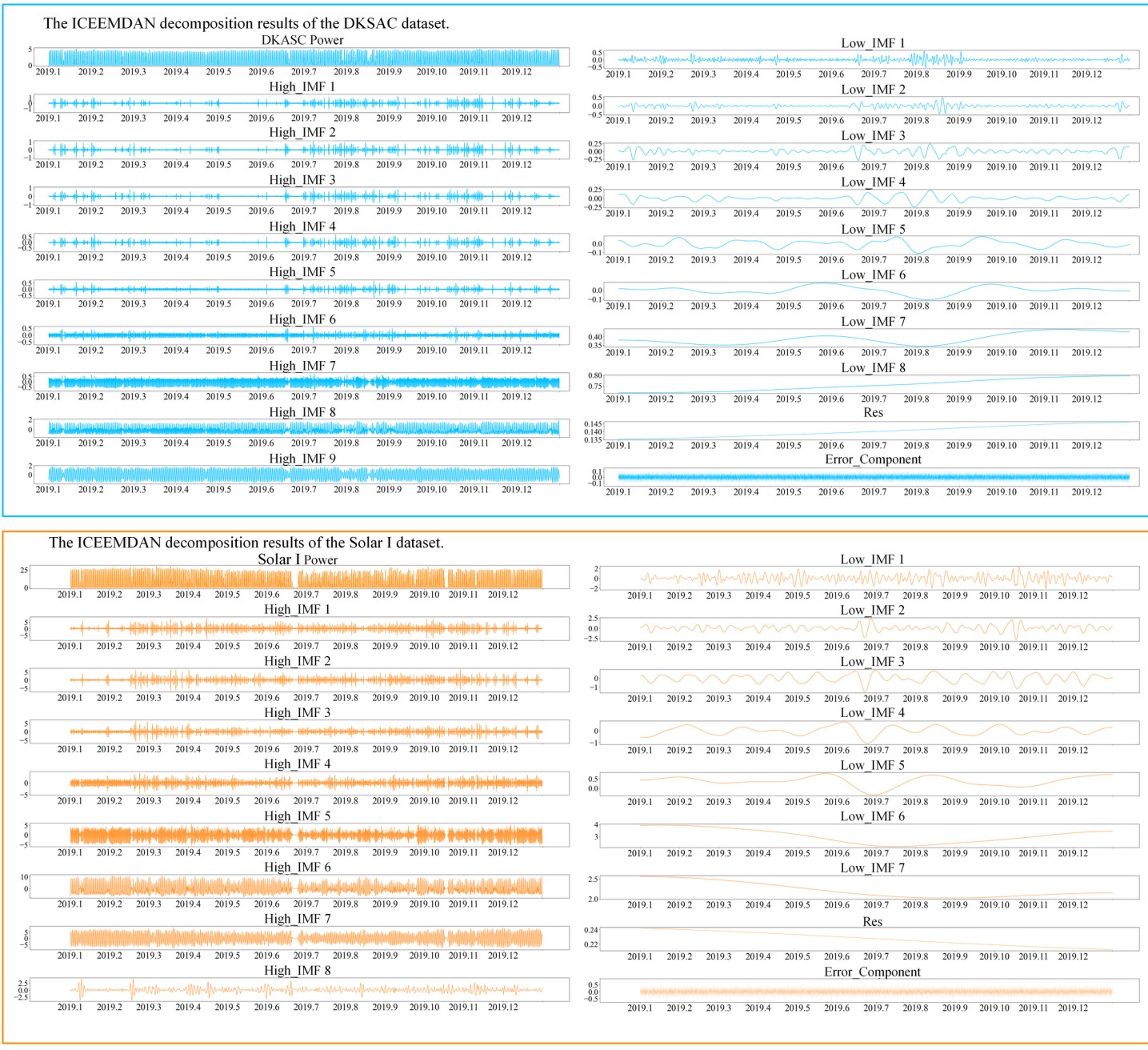

**Fig 5**. **Power sequence decomposition of dataset.**

based on whether the zero-crossing rate is greater than or equal to 0.02 [52]. The remaining component of the original PV power sequence that is not involved in decomposition is defined as the error component [53]. The high-frequency sub-sequences generally have higher volatility, resulting in sudden changes in the original PV power sequence. In contrast, the low-frequency sub-sequences, which change more smoothly, exhibit periodicity in the original PV power sequence. The residual component shows a monotonic curve, and the amplitude of the residual component in 2019 is about 0.0025

kW. The error component usually shows a high frequency of curve fluctuations, but in this study, the error component that is not involved in decomposition fluctuates between -0.1 kW and 0.1 kW, which fully proves the effectiveness of the ICEEMDAN method in decomposing signals, capturing the main components, and reducing the impact of useless noise.

$$Z_i = \frac{n_i}{N_i} \tag{22}$$

Where $Z_i$, $n_i$, and $N_i$ are the zero crossing rates of the $i$th decomposition sequence, the frequency by which the sequence of data crosses the zero threshold, and the total number of data points, respectively.

The study also employs the ICEEMDAN algorithm in order to decompose the PV power series in the Solar I dataset, resulting in the generation of 16 IMFs and 1 Res. As shown in Fig 5, illustrates the disaggregation of the PV power series in the 2019 SolarI dataset, which comprises 35,040 data points. In accordance with the over-zero rate criterion of 0.02, the decomposition results comprise eight groups of high-frequency sub-sequences, seven groups of low frequency sub-sequences, and a group of residual sequences. Residual sequence amplitude in 2019 is approximately 0.005 kW, while the error component amplitude ranges from -0.5 kW to 0.5 kW. The findings indicate that the ICEEMDAN approach maintains excellent decomposition efficacy even when applied to sequences containing anomalous meteorological data.

## 4.4 parameter settings

Hyperparameter tuning is essential for achieving optimal model performance; however, an exhaustive search can be computationally prohibitive. To balance efficiency with effectiveness and enhance experimental reproducibility, this study employed a two-stage optimization strategy. First, an evidence-based initial search space for the core hyperparameters was established through a systematic review of literature in recent years on photovoltaic forecasting [54–57]. Subsequently, within this empirically defined range, the PSO algorithm was utilized for efficient automated tuning to determine the optimal parameter combination for the quantile regression interval forecasting task. The resulting hyperparameters are detailed in Table 4.

Following a series of experimental trials, the number of epochs was set to 100. The batch size was 64 and Adam was selected as the optimizer. Initially, the learning rate was set to 0.001 and then it gradually decayed.

## 4.5 Model training

Ablation experiments were conducted on the ICIAL model .The experimental outcomes were compared. Furthermore, the benchmark models (including LSTM, TCN [11], Informer [58], Itransformer and DA-GRU ) and the ICIAL model proposed in this paper were subjected to training and prediction experiments on the DKASC and Solar I datasets.Then, the evaluation metrics of the benchmark and ICIAL models under various weather conditions are compared. Additionally, the evaluation metrics and visualisation results for these models in a single-step and multi-step prediction task are presented, highlighting the differences in prediction accuracy among the benchmark and ICIAL models.

**Table 4**. Model Hyperparameters.

| Model | Hyperparameter | Range | Optimal Value |
|-------|----------------|-------|---------------|
| CNN | filters | (30, 260) | 128 |
| | kernel_size | (1, 4) | 3 |
| LSTM | layers | (1, 4) | 3 |
| | units | (30, 260) | 128 |
| BiGRU | layers | (1, 4) | 2 |
| | units | (30, 260) | 128 |

The hardware environment utilised in this experiment comprises Intel Core i9-13980HX CPU and 16GB RAM. The software environment is as follows: The operating system utilized in this experiment is Windows 11, with Python 3.9.18 and Pytorch 2.0.0 as the respective programming languages.

## 5 Evaluation

The goal of this section is to evaluate the prediction performance, robustness, and uncertainty quantification of each model for practical applications. The evaluation begins with ablation experiments to assess the contributions of the ICIAL model's individual components. Next, the ICIAL model is benchmarked against baselines for single-step prediction accuracy, with further tests assessing its adaptability under varied weather conditions. This is followed by multi-step prediction experiments across different datasets and strategies to explore the model's long-horizon forecasting capability. To ensure the statistical reliability of these findings, Wilcoxon signed-rank tests are employed to verify the model's robustness against the randomness inherent in deep learning and PSO algorithms. Finally, to overcome the limitations of traditional point predictions, the study introduces uncertainty quantification by generating and evaluating probabilistic prediction intervals under various weather conditions, assessing their reliability (PICP) and sharpness (PIAW) to demonstrate the model's practical capabilities.

### 5.1 Evaluation metrics of model performance

The experiment employed various evaluation metrics to assess the predictive performance of different models. The formulas for calculating MAE, nMAE, RMSE, nRMSE, and SMAPE are as follows [59]:

$$MAE = \frac{1}{n} \sum_{i=1}^{n} |y_i - \hat{y}_i| \tag{23}$$

$$nMAE = \frac{1}{n} \sum_{i=1}^{n} \frac{|y_i - \hat{y}_i|}{\bar{y}} \tag{24}$$

$$RMSE = \sqrt{\frac{1}{n} \sum_{i=1}^{n} (y_i - \hat{y}_i)^2} \tag{25}$$

$$nRMSE = \frac{\sqrt{\frac{1}{n} \sum_{i=1}^{n} (y_i - \hat{y}_i)^2}}{max(y) - min(y)} \tag{26}$$

$$SMAPE = \frac{1}{n} \sum_{i=1}^{n} \frac{|y_i - \hat{y}_i|}{(|y_i| + |\hat{y}_i| + 0.1)/2} \tag{27}$$

where $y_i$, $\hat{y}_i$, and $\bar{y}$ represent the true value, predicted value, and mean of the true values, respectively, while $n$ denotes the total number of observations. Lower values of MAE, nMAE, RMSE, nRMSE, and SMAPE indicate higher accuracy. MAE and RMSE measure the average error and the root mean squared error, respectively. nMAE normalizes MAE by the mean of the true values, whereas nRMSE utilizes the range of the true values, facilitating comparisons across different scales. SMAPE, a symmetric percentage error, is less sensitive to extreme values and remains stable with a bias of 0.1 when both the true and predicted values are zero.

## 5.2 Evaluation on DKASC dataset

The ICIAL model proposed in this paper was evaluated through a series of experiments conducted on the DKASC dataset, comparing its performance against several benchmark models. These experiments included ablation studies, single-step ahead forecasting, single-step ahead forecasting under varying weather conditions, and multi-step ahead forecasting. In the multi-step ahead forecasting experiments, three distinct strategies were employed to address the prediction tasks: the recursive strategy, the direct strategy, and the MIMO strategy. A comparative analysis of experimental metrics and visualization results across the three forecasting strategies demonstrates that the proposed ICIAL model significantly outperforms other benchmark models in all forecasting tasks, confirming its greater capability in PV power forecasting.

**5.2.1 Ablation experiment.** Ablation experiments were conducted on the ICIAL model, and comparisons were made between the result. The model was trained in accordance with the experimental environment and settings described above, as well as the previously determined optimal hyperparameters. Subsequently, the trained models were employed to predict the test-sets. The predicted outcomes were compared to the actual values, and the results were analyzed in detail. Table 5 illustrates the predictive capacity of each model with respect to the dataset.

Based on the results presented in Table 5, the traditional BiGRU model lacks a mechanism to account for the varying importance of time steps, which leads to suboptimal performance. In contrast, the ConvBiGRU model enhances predictive accuracy, reducing MAE by 1.8% compared to BiGRU. The ConvBiGRU-A model, which includes an additional attention mechanism, further decreases the MAE by 4.1% relative to BiGRU. The ConvBiGRU-PE-A model, incorporating position encoding, achieves an 8.6% MAE reduction, while the ConvBiGRU-RPE-A model, which utilizes relative position encoding, reduces the MAE by 11.9% compared to BiGRU. The ICIAL model, which employs the ICEEMDAN algorithm and the IRPE mechanism, demonstrates the best performance, achieving a 20.7% reduction in MAE compared to BiGRU. This improvement stems from the model's capacity to capture complex data relationships and learn effective feature representations.

The $u/v$ parameter scaling in IRPE significantly enhances the model's focusing capability under relatively stable conditions. As shown in the attention heatmaps for the Average, Peak, and Trough scenarios (Fig 6), the ICIAL model exhibits sharper and brighter diagonal patterns compared to the ConvBiGRU-RPE-A model, which uses standard RPE. This indicates that while standard RPE applies static attention to all relative positions, the $u/v$ scaling mechanism in ICIAL enables adaptive weighting during training. This mechanism amplifies critical periodic dependencies, enhances the perception of peak-trough dynamics, and suppresses irrelevant noise, thereby improving prediction accuracy.

Furthermore, the proposed mechanism demonstrates robust dynamic adaptability, particularly in the Ramp scenario. When faced with rapid photovoltaic power fluctuations, the standard RPE model maintained a fixed periodic attention pattern similar to other scenarios. In contrast, the ICIAL model adapted dynamically: while preserving key periodic memories through sub-diagonals, it substantially intensified the brightness of the main diagonal. This demonstrates that the model can shift its attentional focus to recent historical data in response to input dynamics, allowing it to accurately capture the "slope" and local trends of the changes.

**Table 5**. Indicators for single-step (5 min) projections.

| Model | nMAE | MAE (kW) | nRMSE | RMSE (kW) |
|---|---|---|---|---|
| BiGRU | 0.0515 | 0.0663 | 0.0375 | 0.1681 |
| ConvBiGRU | 0.0506 | 0.0651 | 0.0366 | 0.1675 |
| ConvBiGRU-A | 0.0494 | 0.0636 | 0.0337 | 0.1660 |
| ConvBiGRU-PE-A | 0.0471 | 0.0606 | 0.0345 | 0.1653 |
| ConvBiGRU-RPE-A | 0.0454 | 0.0584 | 0.0339 | 0.1630 |
| **ICIAL** | **0.0408** | **0.0526** | **0.0318** | **0.1569** |

Note: Optimal values for each indicator are bolded.

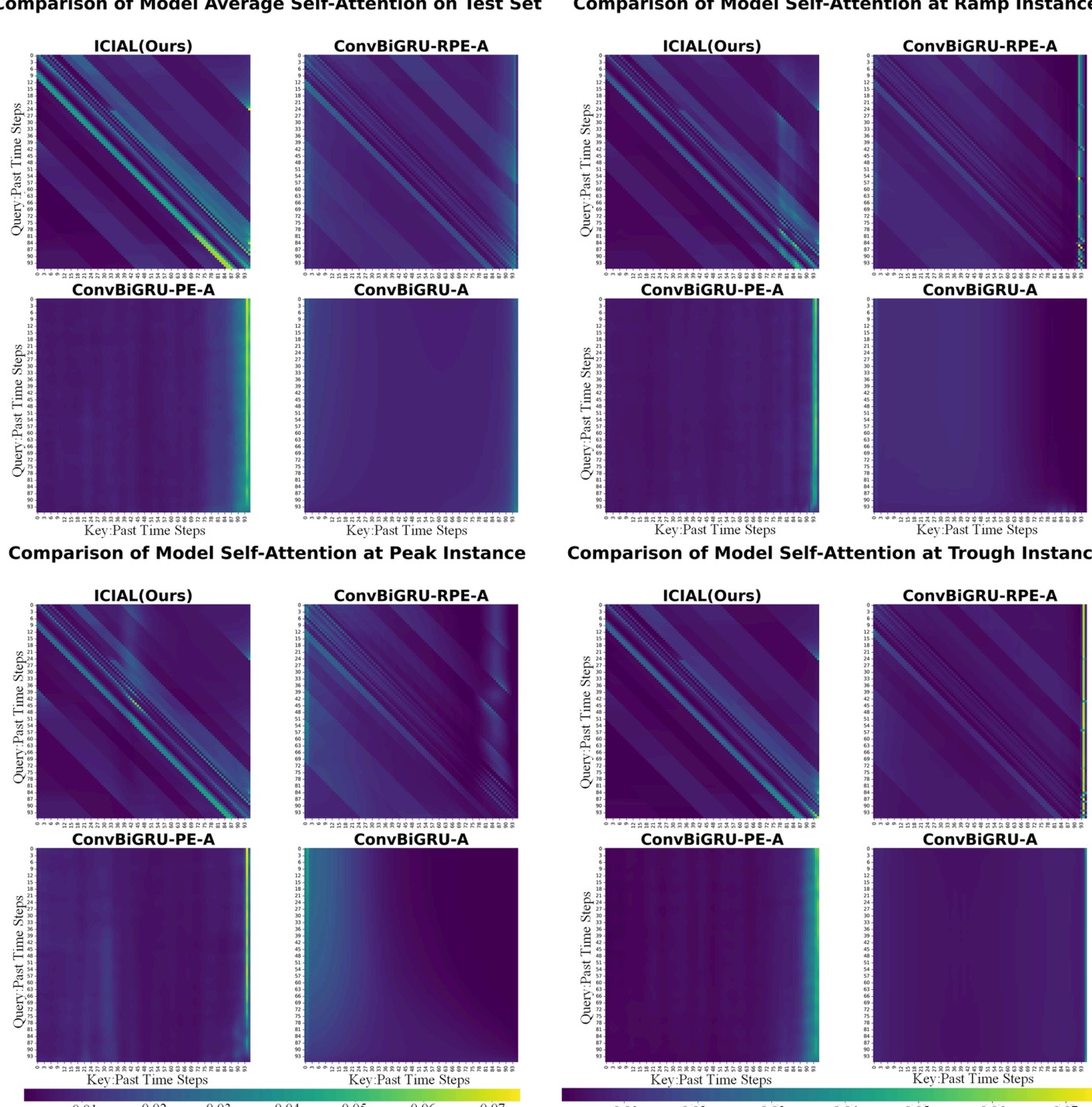

**Fig 6**. Comparison of attention heatmaps between the proposed ICIAL model and the baseline model across four typical scenarios.

### 5.2.2 One-step ahead forecasting.

The ICIAL model was compared with benchmark models in a one-step-ahead forecasting experiment. Fig 7 illustrates the prediction results and the corresponding scatter distributions for each model.

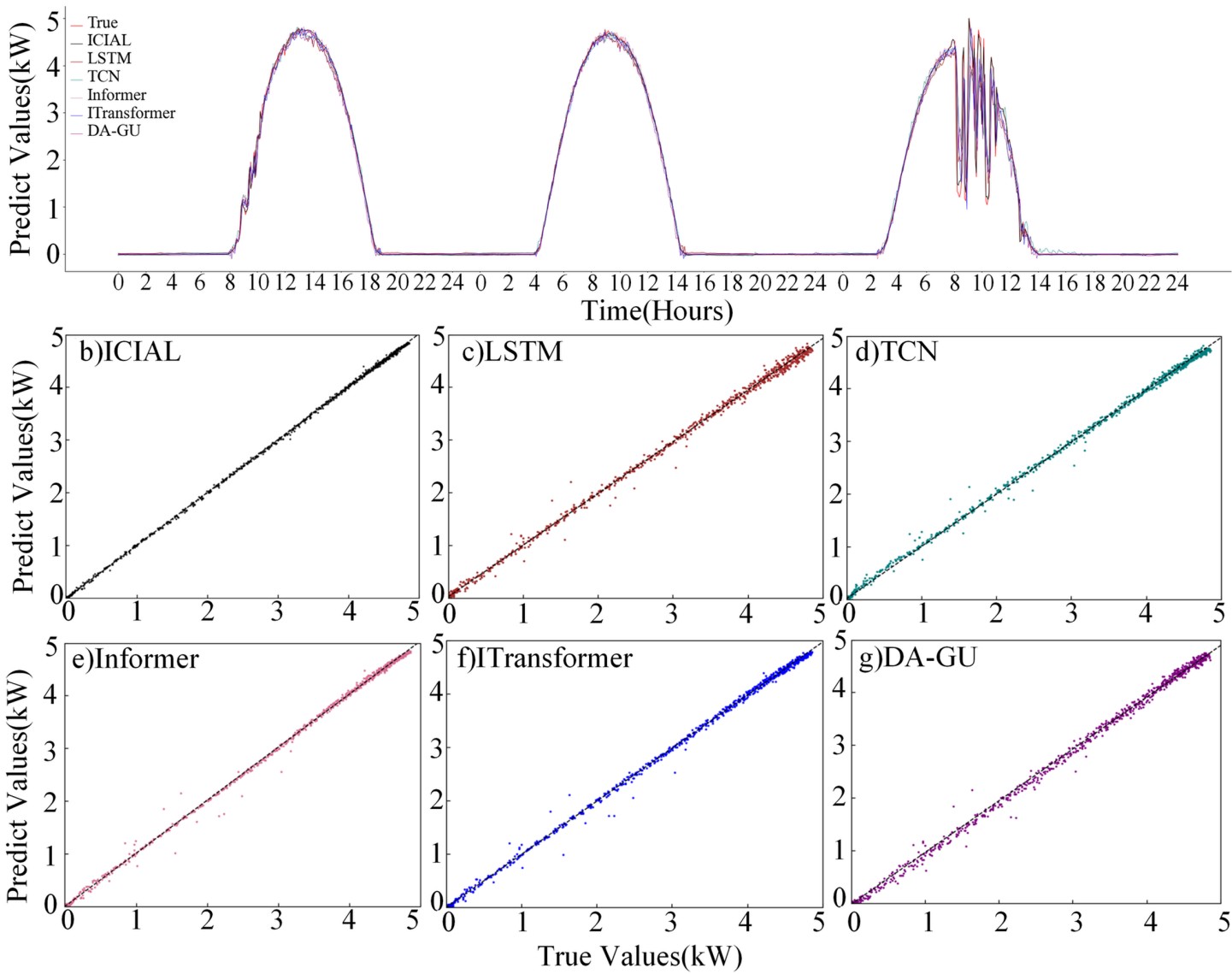

**Fig 7. Advance single-step comparison chart.**

Table 6 presents the MAE, nMAE, RMSE, nRMSE and SMAPE values for the various models in the one-step-ahead prediction.

The ICIAL model demonstrates the highest prediction accuracy among all benchmark models, and its prediction curve closely aligns with the true value of photovoltaic power (see Fig 7(a)). The scatter plots in Fig 7(b)–7(f) illustrate the ICIAL model's high accuracy, as its predicted values form the most compact cluster compared to all other models. Specifically, as detailed in Table 6, ICIAL has made significant improvements in all key metrics when compared to the second-best performing model, DA-GRU. ICIAL's nMAE is 0.0408, which is 4.7% lower than DA-GRU's nMAE of 0.0428. Additionally, its MAE is 0.0526 kW, representing a 4.5% reduction compared to DA-GRU's MAE of 0.0551 kW. Furthermore, the nRMSE and RMSE were also decreased by 2.8% and 1.9%, respectively.

**Table 6**. Indicators for single-step (5 min) projections.

| Model | nMAE | MAE (kW) | nRMSE | RMSE (kW) |
|---|---|---|---|---|
| **ICIAL** | **0.0408** | **0.0526** | **0.0318** | **0.1569** |
| LSTM | 0.0450 | 0.0579 | 0.0350 | 0.1706 |
| TCN | 0.0443 | 0.0570 | 0.0343 | 0.1651 |
| Informer | 0.0440 | 0.0566 | 0.0336 | 0.1635 |
| ITransformer | 0.0447 | 0.0575 | 0.0345 | 0.1699 |
| DA-GRU | 0.0428 | 0.0551 | 0.0327 | 0.1599 |

Note: Optimal values for each indicator are bolded.

Furthermore, the comparison with the worst-performing model, LSTM, underscores the robustness of the ICIAL model. The data presented in Table 6 and the visualization results in Fig 7(a) demonstrate that LSTM's predictions exhibit significant deviations from the actual values during periods of high volatility. In contrast, ICIAL's nMAE and RMSE are 9.3% and 8.0% lower than those of LSTM, respectively. This indicates that LSTM has a limited capacity to capture complex, high-dimensional temporal dependencies, whereas ICIAL's architectural design allows it to adapt more effectively to rapid changes, thereby significantly reducing prediction errors.

To thoroughly evaluate the predictive performance of the ICIAL model in real-world scenarios, comparative experiments were conducted under three different lighting conditions: gradual lighting, fluctuating lighting, and intense lighting.

As shown in Table 7, the prediction accuracy of the ICIAL model surpasses that of all benchmark models under stable, fluctuating, and highly variable lighting conditions. Notably, under highly variable conditions characterized by significant lighting fluctuations, the performance enhancement of the ICIAL model is particularly pronounced, with its MAE and RMSE optimized by approximately 2.3% and 3.5%, respectively, compared to the LSTM model. Among the benchmark models, the DA-GRU model demonstrates relatively strong performance under fluctuating conditions, outperforming all other models except for ICIAL, with an MAE of 0.0249 kW, representing a 50.1% reduction compared to the ITransformer model. However, the overall predictive performance of the DA-GRU model remains inferior to that of the ICIAL model.

**Table 7**. Comparison of metrics for multiple models in three weather conditions.

| Weather Condition | Model | MAE(kW) | RMSE(kW) |
|---|---|---|---|
| Stable | **ICIAL** | **0.0163** | **0.0220** |
| | LSTM | 0.0193 | 0.0395 |
| | TCN | 0.0201 | 0.0454 |
| | Informer | 0.0316 | 0.0467 |
| | ITransformer | 0.0454 | 0.0637 |
| | DA-GRU | 0.0189 | 0.0343 |
| Fluctuating | **ICIAL** | **0.0235** | **0.0366** |
| | LSTM | 0.0254 | 0.0453 |
| | TCN | 0.0266 | 0.0462 |
| | Informer | 0.0333 | 0.0517 |
| | ITransformer | 0.0499 | 0.0694 |
| | DA-GRU | 0.0249 | 0.0439 |
| Volatile | **ICIAL** | **0.1242** | **0.2306** |
| | LSTM | 0.1271 | 0.2390 |
| | TCN | 0.1290 | 0.2414 |
| | Informer | 0.1263 | 0.2373 |
| | ITransformer | 0.1279 | 0.2403 |
| | DA-GRU | 0.1253 | 0.2351 |

Note: Optimal values for each indicator are bolded.

This discrepancy primarily arises from the DA-GRU model's limited adaptability to rapid fluctuations, inadequate extraction of short-term variational features, and its tendency to overfit long-term trends.

In contrast, the outstanding performance of the ICIAL model under these complex conditions can be primarily attributed to its synergistic high- and low-frequency prediction model, as well as the IRPE mechanism integrated into its architecture. These innovations allow the ICIAL model to more effectively capture the intricate multi-dimensional features of photo-voltaic power data and adapt to its inherent variability and non-linear dynamics, thereby significantly enhancing prediction accuracy.

**5.2.3 Multi-step ahead forecasting.** In smart grids, multi-step forecasting is essential for optimizing renewable energy use, maintaining grid stability, and conducting reliable electricity market bidding [12]. To provide a more compre-hensive performance assessment, this study introduces a method of uncertainty quantification based on quantile regres-sion as an alternative to traditional point forecast comparisons. The median of the quantile regression (50th percentile) is utilised as the new point forecast value [42]. This value is then benchmarked against baseline models under multi-step forecasting horizons using recursive, direct, and MIMO multi-step forecasting strategies. This approach enables us to evaluate not only predictive accuracy but also forecast uncertainty, providing a more robust assessment of a model's long-term forecasting capabilities across various strategies.

As indicated in Table 8, the MIMO strategy consistently outperforms both the Direct and Recursive strategies across all models. Notably, the ICIAL model, when paired with the MIMO strategy, achieved the highest performance across all fore-cast horizons. For instance, in the most challenging T+24 long-term forecast, ICIAL's nMAE (0.1142) and RMSE (0.3382 kW) are 10.2% and 17.3% lower, respectively, than those of the runner-up model, ITransformer, demonstrating greater predictive accuracy. Although some models, such as LSTM, show a slight advantage with the Direct strategy in short-term (T+3) forecasts, the performance advantage of the MIMO strategy become more pronounced over longer horizons, highlighting its greater stability and accuracy.

From a computational cost perspective, the Direct strategy is exceedingly expensive due to its "one model per step" mechanism. For the T+24 task, the ICIAL model under the Direct strategy requires four times the number of parame-ters and 3.9 times the training time compared to its MIMO counterpart. Although ICIAL's intrinsic computational overhead (7.13 MB of parameters and approximately 39 minutes of training) is not the lowest, it is comparable to other advanced Transformer models (e.g., Informer), achieving an effective balance between acceptable costs and significant perfor-mance gains.

The results also reveal significant compatibility issues between the model and strategy. The poor performance of ITransformer and TCN under the Recursive strategy stems from a fundamental conflict between their architectures—such as ITransformer's inversion mechanism and TCN's reliance on parallel convolution across complete sequences—and the strategy's iterative, point-by-point mechanism. This finding highlights the importance of aligning model architecture with the forecasting paradigm.

The data not only confirms the high accuracy of the MIMO-ICIAL combination but also highlights the intrinsic robust-ness of the ICIAL model itself. A key observation is the stark performance divergence among models under the error-prone recursive strategy. In contrast to the performance collapse of the ITransformer and the significant decline of the LSTM, ICIAL's performance degradation is far more controlled and gradual (T+24 nMAE: 0.2004). This suggests that ICIAL's unique decomposition and hierarchical architecture provide a stronger inherent capability to suppress noise and error propagation.

Furthermore, an evaluation of the ICIAL model's performance under different multi-step forecasting strategies, as detailed in Fig 8 and Table 8, reveals its robust stability. Among all strategies, the MIMO strategy exhibited the highest long-term stability: from T+3 to T+24, it demonstrated the smallest increases in nMAE (0.0606) and nRMSE (0.0268), while its absolute error values consistently remained the lowest. The Direct strategy followed with moderate stability (nMAE/nRMSE increases of 0.0745/0.0393), whereas the Recursive strategy resulted in the largest final absolute errors due to error accumulation. The heatmap in Fig 8 visually corroborates these performance disparities. Under the MIMO

**Table 8**. Advanced multi-step prediction experiments for the DKASC dataset.

| Time Steps | Model | Strategy | nMAE | MAE(kW) | nRMSE | RMSE(kW) | SMAPE | Params(MB) | Training Time |
|---|---|---|---|---|---|---|---|---|---|
| T+3(15min) | ICAL | MIMO | **0.0536** | **0.0690** | **0.0418** | **0.2061** | **0.09** | 7.13 | 39min |
| | | Direct | 0.0563 | 0.0698 | 0.0430 | 0.2127 | 0.13 | 29.00 | 175min |
| | | Recursive | 0.1271 | 0.1637 | 0.0627 | 0.3088 | 0.39 | 7.13 | 63min |
| | LSTM | MIMO | 0.0569 | 0.0733 | 0.0442 | 0.2179 | 0.11 | **1.32** | **15min** |
| | | Direct | 0.0556 | 0.0725 | 0.0433 | 0.2170 | 0.13 | 5.28 | 61min |
| | | Recursive | 0.1500 | 0.1644 | 0.0704 | 0.4015 | 0.42 | **1.32** | 17min |
| | TCN | MIMO | 0.0693 | 0.0893 | 0.0467 | 0.2303 | 0.11 | 1.71 | 25min |
| | | Direct | 0.0671 | 0.0864 | 0.0466 | 0.2295 | 0.12 | 6.84 | 64min |
| | | Recursive | 0.0731 | 0.0941 | 0.0490 | 0.2413 | 0.17 | 1.71 | 23min |
| | Informer | MIMO | 0.0590 | 0.0759 | 0.0421 | 0.2075 | 0.19 | 9.23 | 50min |
| | | Direct | 0.0644 | 0.0829 | 0.0424 | 0.2090 | 0.17 | 36.92 | 210min |
| | | Recursive | 0.0686 | 0.0883 | 0.0448 | 0.2209 | 0.11 | 9.23 | 65min |
| | ITransformer | MIMO | 0.0569 | 0.0733 | 0.0425 | 0.2096 | 0.11 | 8.13 | 46min |
| | | Direct | 0.0552 | 0.0710 | 0.0420 | 0.2070 | 0.12 | 32.52 | 180min |
| | | Recursive | 0.3817 | 0.4914 | 0.1777 | 0.8757 | 0.34 | 8.13 | 53min |
| | DA-GRU | MIMO | 0.0570 | 0.0734 | 0.0430 | 0.2102 | 0.13 | 3.3 | 18min |
| | | Direct | 0.0549 | 0.0706 | 0.0442 | 0.2180 | 0.15 | 13.2 | 50min |
| | | Recursive | 0.0544 | 0.0700 | 0.0442 | 0.2179 | 0.15 | 3.3 | 29min |
| T+6(30min) | ICAL | MIMO | **0.0646** | **0.0832** | **0.0483** | **0.2383** | **0.12** | 7.13 | 39min |
| | | Direct | 0.0695 | 0.0866 | 0.0491 | 0.2473 | 0.13 | 29.00 | 175min |
| | | Recursive | 0.1201 | 0.1546 | 0.0607 | 0.2992 | 0.32 | 7.13 | 63min |
| | LSTM | MIMO | 0.0665 | 0.0856 | 0.0506 | 0.2495 | 0.12 | **1.32** | **15min** |
| | | Direct | 0.0683 | 0.0904 | 0.0538 | 0.2501 | 0.14 | 5.28 | 61min |
| | | Recursive | 0.1711 | 0.2122 | 0.1033 | 0.4344 | 0.46 | **1.32** | 17min |
| | TCN | MIMO | 0.0991 | 0.1276 | 0.0597 | 0.2942 | 0.12 | 1.71 | 25min |
| | | Direct | 0.0990 | 0.1274 | 0.0606 | 0.2989 | 0.13 | 6.84 | 64min |
| | | Recursive | 0.1300 | 0.1674 | 0.0748 | 0.3687 | 0.19 | 1.71 | 23min |
| | Informer | MIMO | 0.0695 | 0.0895 | 0.0486 | 0.2400 | 0.24 | 9.23 | 50min |
| | | Direct | 0.0691 | 0.0890 | 0.0490 | 0.2420 | 0.18 | 36.92 | 210min |
| | | Recursive | 0.0943 | 0.1213 | 0.0561 | 0.2766 | 0.15 | 9.23 | 65min |
| | ITransformer | MIMO | 0.0657 | 0.0841 | 0.0497 | 0.2403 | 0.12 | 8.13 | 46min |
| | | Direct | 0.0649 | 0.0839 | 0.0489 | 0.2401 | 0.13 | 32.52 | 180min |
| | | Recursive | 0.7612 | 0.9800 | 0.3284 | 1.6185 | 0.60 | 8.13 | 53min |
| | DA-GRU | MIMO | 0.0648 | 0.0836 | 0.0486 | 0.2395 | 0.15 | 3.3 | 18min |
| | | Direct | 0.0660 | 0.0853 | 0.0576 | 0.2547 | 0.16 | 13.2 | 50min |
| | | Recursive | 0.0678 | 0.0873 | 0.0505 | 0.2491 | 0.17 | 3.3 | 29min |
| T+12(60min) | ICAL | MIMO | **0.0784** | **0.1009** | **0.0555** | **0.2737** | **0.15** | 7.13 | 39min |
| | | Direct | 0.0797 | 0.1026 | 0.0564 | 0.2779 | 0.16 | 29.00 | 175min |
| | | Recursive | 0.1336 | 0.1720 | 0.0707 | 0.3484 | 0.32 | 7.13 | 63min |
| | LSTM | MIMO | 0.0867 | 0.1116 | 0.0599 | 0.2951 | 0.16 | **1.32** | **15min** |
| | | Direct | 0.0903 | 0.1170 | 0.0601 | 0.3030 | 0.16 | 5.28 | 61min |
| | | Recursive | 0.1903 | 0.2304 | 0.1505 | 0.4673 | 0.51 | **1.32** | 17min |
| | TCN | MIMO | 0.1606 | 0.2067 | 0.0884 | 0.4355 | 0.19 | 1.71 | 25min |
| | | Direct | 0.1585 | 0.2040 | 0.0888 | 0.4378 | 0.20 | 6.84 | 63min |
| | | Recursive | 0.2777 | 0.3574 | 0.1396 | 0.6878 | 0.28 | 1.71 | 23min |
| | Informer | MIMO | 0.0795 | 0.1024 | 0.0580 | 0.2860 | 0.17 | 9.23 | 50min |
| | | Direct | 0.0917 | 0.1180 | 0.0568 | 0.2798 | 0.18 | 36.92 | 210min |
| | | Recursive | 0.1475 | 0.1899 | 0.0792 | 0.3902 | 0.23 | 9.23 | 65min |
| | ITransformer | MIMO | 0.0803 | 0.1054 | 0.0610 | 0.2926 | 0.16 | 8.13 | 46min |
| | | Direct | 0.0896 | 0.1153 | 0.0611 | 0.3011 | 0.16 | 32.52 | 180min |
| | | Recursive | 0.9349 | 1.2035 | 0.4062 | 2.0021 | 0.76 | 8.13 | 53min |
| | DA-GRU | MIMO | 0.0822 | 0.1058 | 0.0558 | 0.2750 | 0.16 | 3.3 | 18min |
| | | Direct | 0.0847 | 0.1091 | 0.0570 | 0.2810 | 0.18 | 13.2 | 50min |
| | | Recursive | 0.0950 | 0.1222 | 0.0620 | 0.3056 | 0.23 | 3.3 | 29min |

*continued*

**Table 8.** Continued

| Time Steps | Model | Strategy | nMAE | MAE(kW) | nRMSE | RMSE(kW) | SMAPE | Params(MB) | Training Time |
|---|---|---|---|---|---|---|---|---|---|
| T+24(120min) | ICAL | MIMO | **0.1142** | **0.1470** | **0.0686** | **0.3382** | **0.23** | 7.13 | 39min |
| | | Direct | 0.1308 | 0.1683 | 0.0823 | 0.4055 | 0.25 | 29.00 | 175min |
| | | Recursive | 0.2004 | 0.2580 | 0.1141 | 0.5625 | 0.41 | 7.13 | 63min |
| | LSTM | MIMO | 0.1456 | 0.1874 | 0.0925 | 0.4561 | 0.25 | **1.32** | **15min** |
| | | Direct | 0.1474 | 0.1897 | 0.0967 | 0.4617 | 0.26 | 5.28 | 61min |
| | | Recursive | 0.2027 | 0.2615 | 0.1684 | 0.6042 | 0.53 | **1.32** | 17min |
| | TCN | MIMO | 0.2959 | 0.3809 | 0.1496 | 0.7375 | 0.33 | 1.71 | 25min |
| | | Direct | 0.3043 | 0.3960 | 0.1536 | 0.7396 | 0.35 | 6.84 | 64min |
| | | Recursive | 0.6214 | 0.8000 | 0.3016 | 1.4864 | 0.50 | 1.71 | 23min |
| | Informer | MIMO | 0.1364 | 0.1756 | 0.0878 | 0.4326 | 0.28 | 9.23 | 50min |
| | | Direct | 0.1421 | 0.1829 | 0.0890 | 0.4768 | 0.30 | 36.92 | 210min |
| | | Recursive | 0.2800 | 0.3605 | 0.1342 | 0.6616 | 0.40 | 9.23 | 65min |
| | ITransformer | MIMO | 0.1272 | 0.1637 | 0.0830 | 0.4090 | 0.24 | 8.13 | 46min |
| | | Direct | 0.1366 | 0.1758 | 0.0874 | 0.4305 | 0.23 | 32.52 | 180min |
| | | Recursive | 0.9376 | 1.2070 | 0.4059 | 2.0008 | 0.81 | 8.13 | 53min |
| | DA-GRU | MIMO | 0.1338 | 0.1723 | 0.0850 | 0.4190 | 0.27 | 3.3 | 18min |
| | | Direct | 0.1382 | 0.1779 | 0.0867 | 0.4273 | 0.25 | 13.2 | 50min |
| | | Recursive | 0.1729 | 0.2226 | 0.1019 | 0.5023 | 0.34 | 3.3 | 29min |

Note: Optimal values for each indicator are bolded; Training Time for the Direct strategy represents the cumulative time to train all independent models for the forecast horizon. For MIMO and Recursive, it is the time to train a single model; Total parameters for the Direct strategy are the sum of all independent models. For MIMO and Recursive, the value shown is for the single model used.

strategy, the predicted points are compactly clustered around the diagonal identity line (ground truth) across all time steps. Although the Direct strategy performs comparably to MIMO in the short term (T+3), its point cloud becomes significantly more dispersed as the forecast horizon extends, revealing a key limitation in its long-range capability. In contrast, the point cloud of the Recursive strategy is the most scattered and exhibits systematic bias resulting from error propagation. More critically, the ICIAL model demonstrates profound intrinsic robustness even under the inherently flawed Recursive strategy. Compared to the baseline models referenced in Table 8, ICIAL's performance degrades more gracefully, avoiding a collapse of the core predictive trend. This behavior highlights the architecture's robust feature extraction and noise resistance, which enables it to maintain relatively stable performance even under such adverse conditions.

Finally, as illustrated in Fig 9, presents a visualisation of the predicted values under representative weather scenarios. Under relatively stable fluctuating weather conditions (e.g., T+6, T+24), ICIAL's prediction curve demonstrates a high degree of fidelity to the actual values (grey shaded area). It particularly excels in peak prediction accuracy, effectively addressing the peak underestimation problem commonly observed in other baseline models, such as LSTM. Conversely, in abrupt weather scenarios with drastic changes in photovoltaic power (e.g., T+3, T+12), ICIAL exhibits an robust ability to capture dynamic trends. Its prediction curve (solid blue line) closely tracks rapid fluctuations and sharp drops in power, showing significantly less prediction lag compared to other models. Furthermore, a critical observation is that as the forecast horizon extends (from T+3 to T+24), ICIAL's prediction curve remains consistently smoother and more stable than those of the other baseline models. This is particularly evident under the Recursive strategy, where ICIAL's output maintains a robust form, while other models exhibit severe noise fluctuations due to error accumulation, once again demonstrating the robustness of its architecture. This robust capability for peak prediction and rapid trend response holds significant practical importance, as it directly contributes to optimising resource allocation, enhancing operational efficiency, and improving adaptability to market volatility.

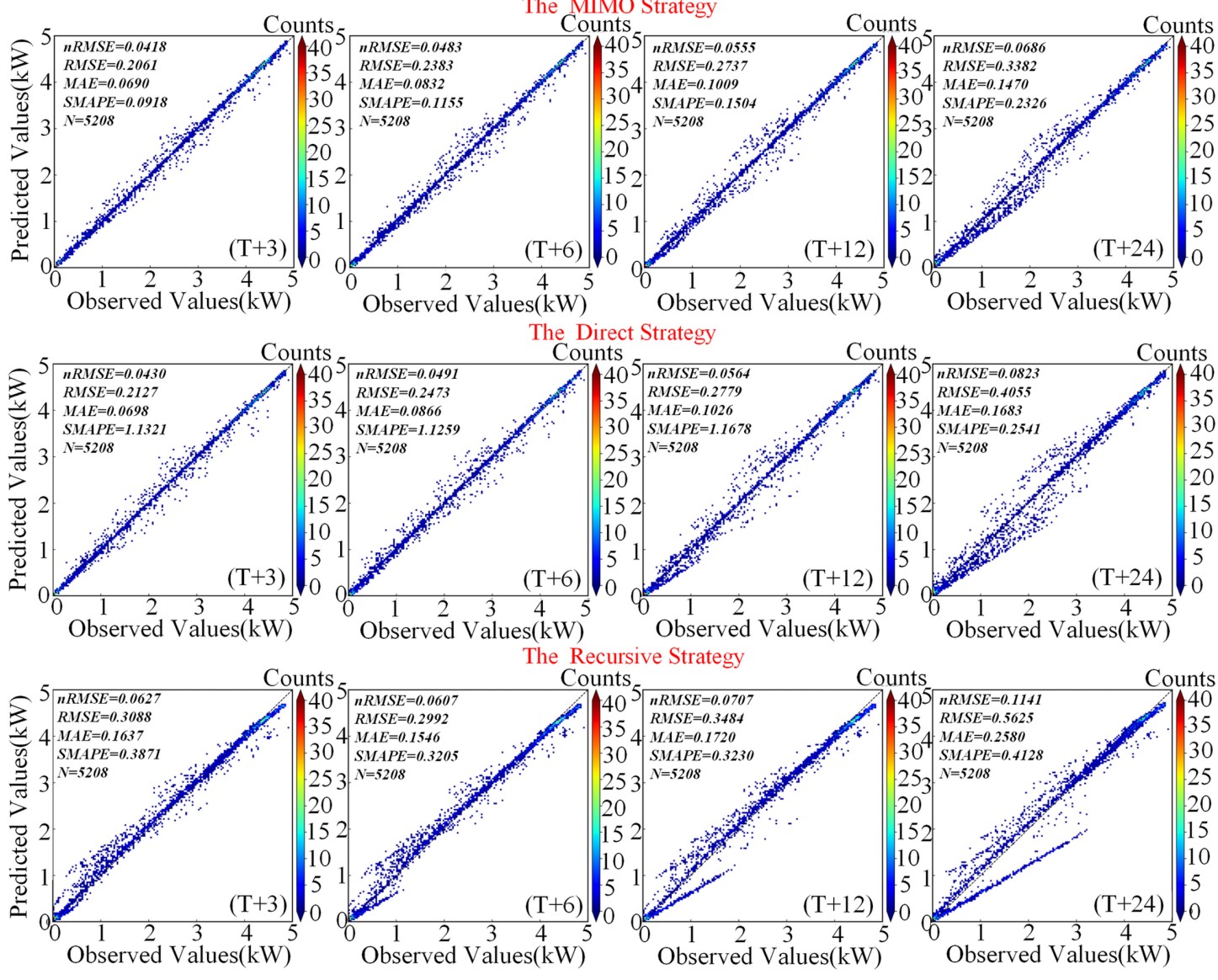

**Fig 8**. Multi-step prediction of heat density in advance under different strategies for the ICAL model.

## 5.3 Evaluation on SolarI dataset.

To mitigate the limitations arising from a single dataset, this study conducts experiments across multiple datasets to enhance the reliability and generalizability of the results. Compared to the DKASC dataset, the Solar I dataset is smaller in scale and exhibits greater volatility, posing more stringent demands on the model's feature extraction and temporal information processing capabilities. Consequently, evaluations on the Solar I dataset provide a more rigorous test of the model's robustness and adaptability.

As illustrated in Table 9, the experimental results clearly reproduce the core conclusions observed in the DKASC dataset: the ICIAL model employing the MIMO strategy remains the best-performing combination, demonstrating strong cross-dataset adaptability. Taking the T+192 (2880-minute) long-term forecast as an example, the ICIAL model's nMAE

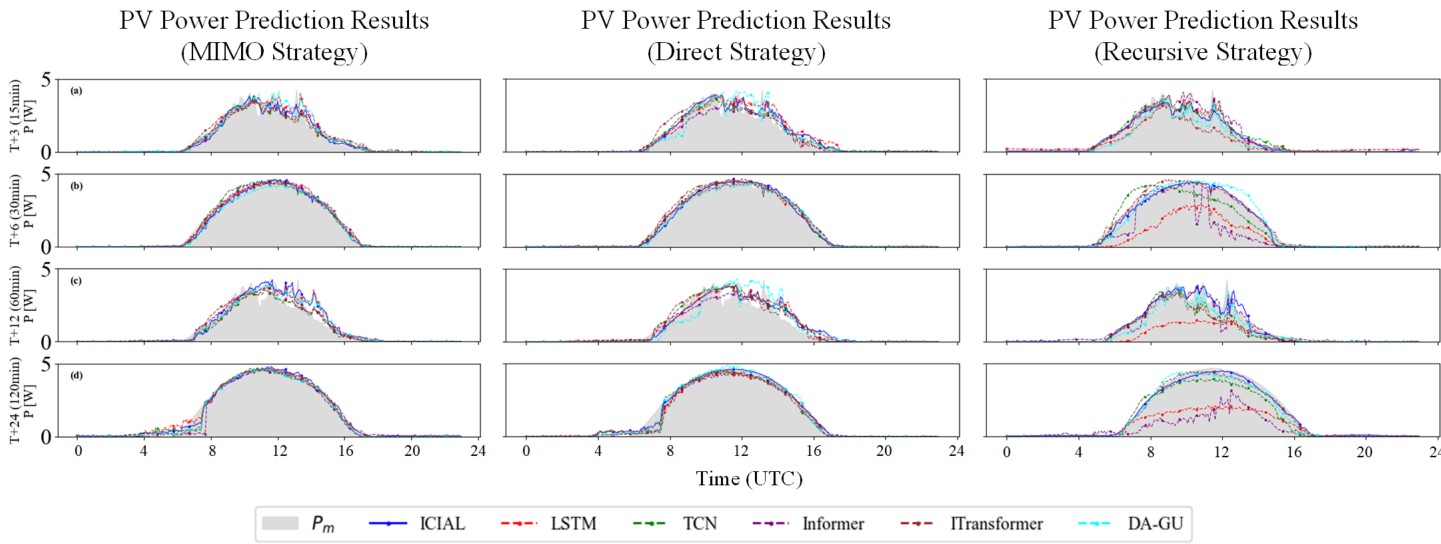

**Fig 9**. **Multi-step ahead prediction plots for each model.**

not only significantly exceeds that of the classic LSTM-Recursive baseline strategy (with a reduction of 61.2%), but its nRMSE also drastically outperforms the worst-performing TCN-Recursive strategy (with a reduction of 68.7%). Furthermore, it surpasses the robustly performing DA-GRU model (MIMO strategy), with nMAE and nRMSE reductions of 8.1% and 5.9%, respectively. This highlights its robust discriminative ability on complex data. In terms of training efficiency, the ICIAL model's training time of approximately 13 minutes under the MIMO strategy ensures the highest accuracy while maintaining computational efficiency comparable to other advanced models.

### 5.4 Experimental evaluation of model stability using Wilcoxon signed-rank test

To rigorously evaluate the stability of the proposed photovoltaic power forecasting model utilizing the MIMO prediction strategy, which integrates deep learning and PSO, a comprehensive experiment was conducted to assess its robustness against the inherent randomness associated with the stochastic nature of both deep learning and PSO. The experiment aimed to quantify the consistency of predictions across multiple runs, employing two distinct datasets—DKASC and Solar I—across various forecasting horizons (T+1, T+3, T+6, T+12, and T+24). The Wilcoxon signed-rank test, a non-parametric statistical method, was utilized to compare prediction results over 30 independent runs, ensuring a thorough assessment of model stability.

The DKASC and Solar I datasets were selected to represent diverse photovoltaic power generation scenarios, with DKASC exhibiting relatively stable patterns and Solar I characterized by higher variability due to environmental factors. For each dataset and forecasting horizon, the model was executed 30 times, and the differences between predicted and true values (Predicted - True) were recorded.The Wilcoxon signed-rank test was applied to evaluate the statistical significance of differences between runs, with p-values calculated for each comparison. The median p-value and the proportion of significant differences ($p < 0.05$) were reported as key indicators of stability. Additionally, box plots and histograms were utilized to visualize the distribution of prediction errors and p-values, respectively, offering a comprehensive perspective on the model's performance and consistency.

The results, presented in Table 10, highlight the model's robust stability across all forecasting horizons and datasets. The median p-values from the Wilcoxon signed-rank test consistently exceed the significance threshold of 0.05, ranging from 0.3626 to 0.6989, indicating that the prediction results across multiple runs are statistically indistinguishable in most

**Table 9**. Advanced multi-step prediction experiments for the solar I dataset.

| Time Steps | Model | Strategy | nMAE | MAE(kW) | nRMSE | RMSE(kW) | SMAPE | Params(MB) | Training Time |
|---|---|---|---|---|---|---|---|---|---|
| T+3(45min) | ICAL | MIMO | **0.1169** | **0.7698** | **0.0654** | **1.7467** | **0.15** | 7.18 | 20min |
| | | Direct | 0.1359 | 0.8928 | 0.0687 | 1.8343 | 0.17 | 28.72 | 73min |
| | | Recursive | 0.1387 | 0.9353 | 0.0761 | 2.0328 | 0.60 | 7.18 | 49min |
| | LSTM | MIMO | 0.1846 | 1.2156 | 0.0810 | 1.9763 | 0.21 | **1.35** | **4min** |
| | | Direct | 0.1428 | 0.9404 | 0.2419 | 1.9924 | 0.53 | 5.4 | 13min |
| | | Recursive | 0.1313 | 0.8649 | 0.1395 | 2.0873 | 0.33 | **1.35** | 7min |
| | TCN | MIMO | 0.1598 | 0.9058 | 0.0766 | 1.9578 | 0.24 | 1.73 | 4min |
| | | Direct | 0.1606 | 0.9572 | 0.0871 | 1.9701 | 0.56 | 6.92 | 22min |
| | | Recursive | 0.1713 | 0.9938 | 0.0951 | 1.9818 | 0.49 | 1.73 | 7min |
| | Informer | MIMO | 0.1365 | 0.8991 | 0.0727 | 1.8915 | 0.16 | 9.25 | 32min |
| | | Direct | 0.1311 | 0.8631 | 0.0705 | 1.8835 | 0.19 | 37 | 180min |
| | | Recursive | 0.1366 | 0.8993 | 0.0731 | 1.9520 | 0.23 | 9.25 | 117min |
| | ITransformer | MIMO | 0.1279 | 0.8590 | 0.0724 | 1.9338 | 0.15 | 8.16 | 9min |
| | | Direct | 0.1260 | 0.8563 | 0.0764 | 2.0395 | 0.16 | 32.64 | 30min |
| | | Recursive | 0.1368 | 0.9223 | 0.0826 | 2.2047 | 0.25 | 8.16 | 9min |
| | DA-GRU | MIMO | 0.1330 | 0.8664 | 0.0740 | 1.9764 | 0.15 | 3.5 | 9min |
| | | Direct | 0.1360 | 0.8773 | 0.0724 | 1.9321 | 0.15 | 14 | 36min |
| | | Recursive | 0.1397 | 0.8933 | 0.0761 | 2.0626 | 0.18 | 3.5 | 10min |
| T+24(360min) | ICAL | MIMO | **0.1356** | **0.8932** | **0.0679** | **1.8132** | **0.16** | 7.18 | 20min |
| | | Direct | 0.1391 | 0.9099 | 0.0735 | 1.9634 | 0.40 | 28.72 | 73min |
| | | Recursive | 0.1566 | 1.0494 | 0.0833 | 2.2234 | 0.73 | 7.18 | 49min |
| | LSTM | MIMO | 0.2908 | 1.3755 | 0.0968 | 2.1036 | 0.31 | **1.35** | **4min** |
| | | Direct | 0.3031 | 1.3898 | 0.1144 | 2.3424 | 0.91 | 5.4 | 13min |
| | | Recursive | 0.3158 | 1.3954 | 0.2054 | 2.3622 | 0.46 | **1.35** | 7min |
| | TCN | MIMO | 0.2708 | 1.0581 | 0.0896 | 2.0778 | 0.28 | 1.73 | 4min |
| | | Direct | 0.2834 | 1.0864 | 0.1511 | 2.1445 | 0.77 | 6.92 | 22min |
| | | Recursive | 0.2990 | 1.1910 | 0.2383 | 2.6676 | 0.74 | 1.73 | 7min |
| | Informer | MIMO | 0.1497 | 0.9081 | 0.0830 | 2.2165 | 0.26 | 9.25 | 32min |
| | | Direct | 0.1512 | 0.9160 | 0.0827 | 2.2071 | 0.31 | 37 | 180min |
| | | Recursive | 0.1701 | 0.9719 | 0.0846 | 2.2595 | 0.25 | 9.25 | 117min |
| | ITransformer | MIMO | 0.1427 | 0.9400 | 0.0763 | 2.0375 | 0.18 | 8.16 | 9min |
| | | Direct | 0.1447 | 0.9531 | 0.0784 | 2.0938 | 0.17 | 32.64 | 30min |
| | | Recursive | 0.1898 | 1.2717 | 0.1057 | 2.8232 | 0.28 | 8.16 | 9min |
| | DA-GRU | MIMO | 0.1399 | 0.8940 | 0.0810 | 2.1646 | 0.18 | 3.5 | 9min |
| | | Direct | 0.1419 | 0.9350 | 0.0800 | 2.1352 | 0.17 | 14 | 36min |
| | | Recursive | 0.1735 | 1.1629 | 0.1015 | 2.7107 | 0.19 | 3.5 | 10min |
| T+96(1440min) | ICAL | MIMO | **0.1412** | **0.9323** | **0.0797** | **2.1269** | **0.16** | 7.18 | 20min |
| | | Direct | 0.1483 | 0.9405 | 0.0803 | 2.1836 | 0.35 | 28.72 | 73min |
| | | Recursive | 0.1622 | 1.0760 | 0.0879 | 2.3459 | 0.80 | 7.18 | 49min |
| | LSTM | MIMO | 0.2991 | 1.4030 | 0.1021 | 2.1321 | 0.22 | **1.35** | **4min** |
| | | Direct | 0.3114 | 1.4506 | 0.1391 | 2.5658 | 0.92 | 5.4 | 13min |
| | | Recursive | 0.3535 | 1.4600 | 0.2204 | 2.5792 | 0.70 | **1.35** | 7min |
| | TCN | MIMO | 0.2831 | 1.1539 | 0.0917 | 2.1422 | 0.29 | 1.73 | 4min |
| | | Direct | 0.3344 | 1.1880 | 0.1786 | 2.2297 | 0.31 | 6.92 | 22min |
| | | Recursive | 0.3567 | 1.2316 | 0.2507 | 2.8735 | 0.92 | 1.73 | 7min |
| | Informer | MIMO | 0.1607 | 0.9936 | 0.0830 | 2.2158 | 0.28 | 9.25 | 32min |
| | | Direct | 0.1583 | 0.9540 | 0.0862 | 2.2991 | 0.35 | 37 | 180min |
| | | Recursive | 0.1951 | 1.0083 | 0.0865 | 2.3072 | 0.36 | 9.25 | 117min |
| | ITransformer | MIMO | 0.1439 | 0.9483 | 0.0809 | 2.1598 | 0.20 | 8.16 | 9min |
| | | Direct | 0.1605 | 1.0565 | 0.0830 | 2.2150 | 0.18 | 32.64 | 30min |
| | | Recursive | 0.2384 | 1.5817 | 0.1168 | 3.1187 | 0.39 | 8.16 | 9min |
| | DA-GRU | MIMO | 0.1475 | 0.9395 | 0.0856 | 2.1664 | 0.19 | 3.5 | 9min |
| | | Direct | 0.1428 | 0.9380 | 0.0831 | 2.1425 | 0.22 | 14 | 36min |
| | | Recursive | 0.2089 | 1.3859 | 0.1234 | 3.2950 | 0.39 | 3.5 | 10min |
| T+192(2880min) | ICAL | MIMO | **0.1490** | **0.9854** | **0.0899** | **2.3999** | **0.17** | 7.31 | 20min |
| | | Direct | 0.1542 | 1.0153 | 0.0990 | 2.6835 | 0.55 | 28.72 | 73min |
| | | Recursive | 0.1684 | 1.1086 | 0.0921 | 2.4588 | 0.83 | 7.31 | 49min |

*continued*

**Table 9**. Continued

| Time Steps | Model | Strategy | nMAE | MAE(kW) | nRMSE | RMSE(kW) | SMAPE | Params(MB) | Training Time |
|---|---|---|---|---|---|---|---|---|---|
| | LSTM | MIMO | 0.3069 | 1.5962 | 0.0914 | 2.4400 | 0.24 | **1.35** | **4min** |
| | | Direct | 0.3551 | 1.6872 | 0.1058 | 2.8246 | 0.97 | 5.4 | 13min |
| | | Recursive | 0.3839 | 1.7966 | 0.1041 | 2.7794 | 0.96 | **1.35** | 7min |
| | TCN | MIMO | 0.2975 | 1.2841 | 0.0906 | 2.4200 | 0.30 | 1.73 | 4min |
| | | Direct | 0.3765 | 1.2571 | 0.2199 | 2.5694 | 0.42 | 6.92 | 22min |
| | | Recursive | 0.3865 | 1.8700 | 0.2875 | 2.9071 | 0.94 | 1.73 | 7min |
| | Informer | MIMO | 0.1721 | 1.3735 | 0.0910 | 2.4300 | 0.31 | 9.25 | 32min |
| | | Direct | 0.1836 | 1.4029 | 0.1190 | 2.3768 | 0.35 | 37 | 180min |
| | | Recursive | 0.2051 | 1.5553 | 0.1203 | 2.3785 | 0.37 | 9.25 | 117min |
| | ITransformer | MIMO | 0.1854 | 1.2212 | 0.0961 | 2.4327 | 0.23 | 8.16 | 9min |
| | | Direct | 0.1635 | 1.0764 | 0.0904 | 2.4129 | 0.28 | 32.64 | 30min |
| | | Recursive | 0.2574 | 1.6952 | 0.1179 | 3.1477 | 0.43 | 8.16 | 9min |
| | DA-GRU | MIMO | 0.1622 | 1.0725 | 0.0955 | 2.4093 | 0.21 | 3.5 | 9min |
| | | Direct | 0.1627 | 1.0759 | 0.0918 | 2.4500 | 0.25 | 14 | 36min |
| | | Recursive | 0.2958 | 1.7178 | 0.1750 | 4.6725 | 0.43 | 3.5 | 10min |

Note: Optimal values for each indicator are bolded; Training Time for the Direct strategy represents the cumulative time to train all independent models for the forecast horizon. For MIMO and Recursive, it is the time to train a single model; Total parameters for the Direct strategy are the sum of all independent models. For MIMO and Recursive, the value shown is for the single model used.

**Table 10**. **Median p-values and significant differences for forecasting time steps.**

| Timestamps | Databases | Median p-value | Proportion of significant differences |
|---|---|---|---|
| T+1 | DKASC | 0.5422 | 0.00% |
| | Solar I | 0.5787 | 0.00% |
| T+3 | DKASC | 0.3626 | 0.00% |
| | Solar I | 0.6989 | 0.00% |
| T+6 | DKASC | 0.4318 | 0.00% |
| | Solar I | 0.4535 | 3.33% |
| T+12 | DKASC | 0.5166 | 0.00% |
| | Solar I | 0.4854 | 6.45% |
| T+24 | DKASC | 0.4768 | 3.33% |
| | Solar I | 0.3695 | 6.67% |

cases. Specifically, for short-term forecasts (T+1 and T+3), the proportion of significant differences ($p < 0.05$) is 0.00% for both the DKASC and Solar I datasets, underscoring the model's robust consistency in these scenarios. For longer forecasting horizons (T+6, T+12, and T+24), the proportion of significant differences remains low. The DKASC dataset exhibits 0.00% significant differences from T+1 to T+12, increasing slightly to 3.33% at T+24, while the Solar I dataset shows a gradual rise to 3.33% at T+6, 6.45% at T+12, and 6.67% at T+24. These findings suggest that the model maintains strong stability even as the forecasting horizon extends, with only a marginal increase in variability observed for the more challenging Solar I dataset.

Fig 10 further validates the model's stability by depicting the distribution of prediction errors across 30 runs. For the DKASC dataset, the error distributions are tightly clustered, ranging from $\pm0.8$kW at T+1 to $\pm1.0$kW at T+24, with median errors consistently near zero, indicating negligible systematic bias. In contrast, the Solar I dataset exhibits slightly greater variability, with error ranges increasing from $\pm0.6$kW at T+1 to $\pm1.5$kW at T+24; however, the distributions remain symmetric and centered around zero, with minimal outliers. This visual analysis underscores the model's high consistency across runs, with the DKASC dataset demonstrating robust stability compared to Solar I.

Histograms of p-values (Fig 11) offer deeper insight into the statistical consistency of the model. For the DKASC dataset, p-values are predominantly distributed above 0.2 across all forecasting horizons, with only a few instances falling

below the 0.05 threshold, aligning with the low proportion of significant differences reported in Table 10. Conversely, the Solar I dataset displays a slightly broader distribution, with a modest increase in p-values below 0.05 at longer horizons (T+12 and T+24), consistent with the observed increase in significant differences. However, even in the most challenging case (Solar I at T+24), the proportion of p-values below 0.05 remains low at 6.67%, indicating that the model's predictions are largely stable across runs.

The experimental evaluation demonstrates that the proposed photovoltaic power forecasting model exhibits high stability across diverse datasets and forecasting horizons, effectively mitigating the randomness introduced by deep learning and PSO algorithms. Results from the Wilcoxon signed-rank test, supported by median p-values consistently exceeding 0.05 and a low proportion of significant differences (maximum of 6.67%), confirm the statistical consistency of the model's predictions across multiple runs. Box plots and p-value histograms further substantiate this finding, revealing tightly clustered error distributions and a predominance of high p-values, respectively. Although the Solar I dataset shows slightly elevated variability in longer-term forecasts, the overall impact of randomness remains minimal, with the model sustaining robust performance. These findings underscore the reliability of the proposed approach for photovoltaic power forecasting, positioning it as a promising solution for real-world applications where stability and consistency are paramount.

## 5.5 Uncertainty quantification and interval prediction under various weather conditions

To address the limitations of deterministic point prediction and enhance the practical utility of models in PV power time-series forecasting, this study conducts uncertainty quantification by generating probabilistic prediction intervals. A quantile regression method is adopted to construct these intervals, with the PICP and PIAW introduced as core evaluation metrics. This section aims to comprehensively and rigorously evaluate the model's uncertainty and interval prediction performance under different weather conditions (including stable, fluctuating, and abrupt weather) from two dimensions: reliability and sharpness.

As shown in Fig 12, the analysis of 95% confidence prediction intervals profoundly reveals the model's robust capability to quantify uncertainties across different datasets and diverse weather conditions. Particularly on the Solar I dataset, where weather conditions are more complex and the prediction time steps are longer (up to 360 minutes), the model demonstrates remarkable robustness. Even when facing the dual challenges of long-term prediction and abrupt weather changes, the generated prediction intervals (shaded areas) effectively envelop the actual power values (black line).

Meanwhile, on the DKASC dataset, the model exhibits similar adaptive capabilities: the interval width increases reasonably as the prediction horizon extends from 15 to 120 minutes. The ability to provide reliable and dynamically adjusted prediction intervals across two datasets with distinct characteristics fully validates the model's strong performance in uncertainty estimation and its practical application value.

As shown in Table 11, the quantitative evaluation of prediction intervals confirms these visual observations. For the DKASC dataset, PICP values exhibit excellent calibration, closely matching the nominal confidence levels. At the 95% confidence level, the PICP values at prediction steps 3, 6, 12, and 24 are 96.56%, 95.89%, 95.70%, and 94.01%, respectively, remaining high and stable, which demonstrates a high degree of reliability. However, despite the greater challenges posed by the more volatile Solar I dataset, especially at the 24-step prediction horizon (PICP of 90.44%), the coverage remains robust, validating the model's effectiveness. The trend of PICP values slightly exceeding their nominal confidence levels reflects a conservative yet reliable model, which is more desirable for risk-averse operational planning in power systems. In terms of sharpness, PIAW values increase reasonably with the extension of the prediction horizon and the rise of confidence levels. For the 5.2 kW system in DKASC, at the 95% confidence level, as the prediction horizon extends from T+3 to T+24, the PIAW increases from 0.5747 kW to 0.7678 kW, indicating that the intervals are narrow enough to be practically valuable.

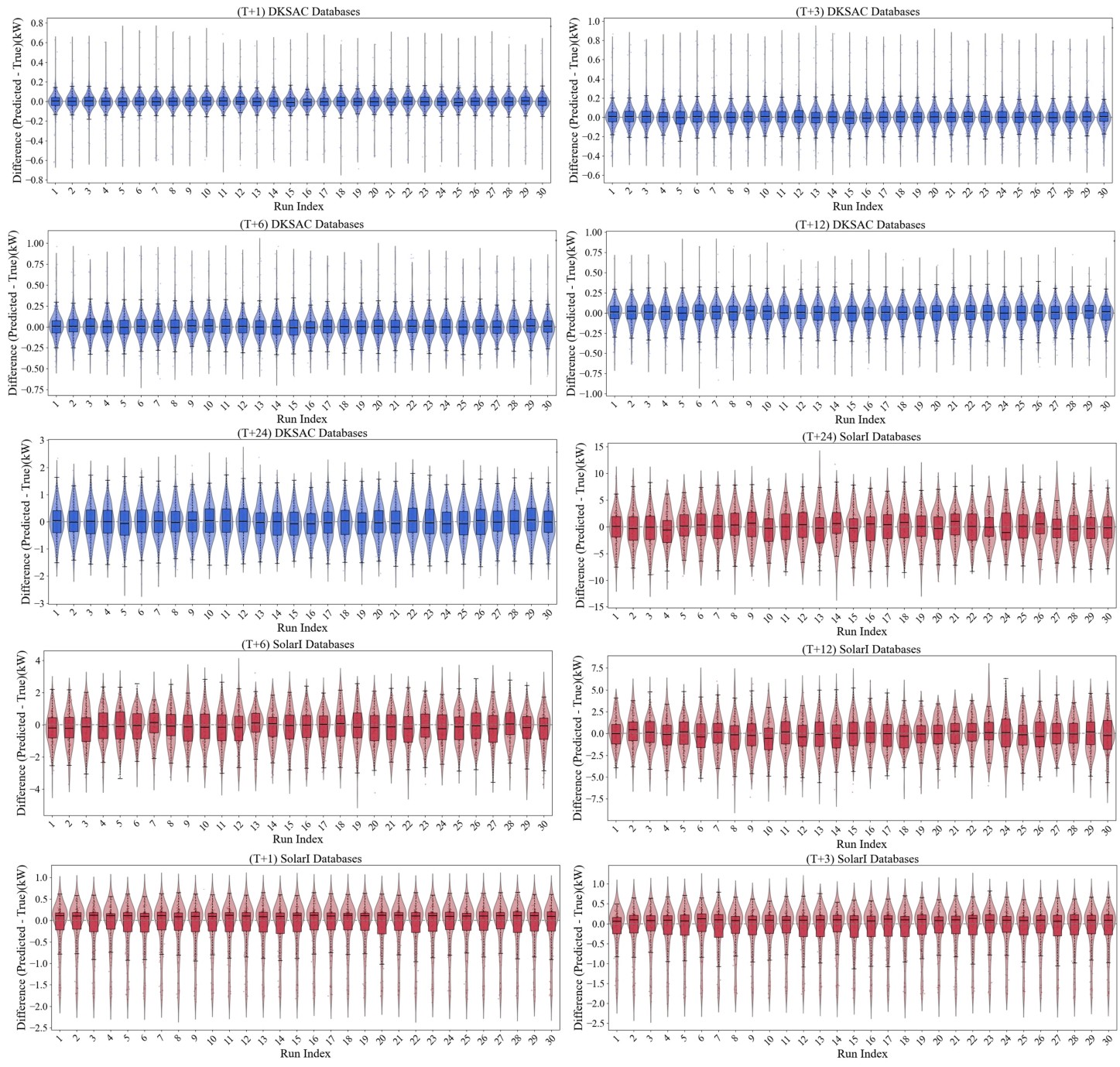

**Fig 10. Distribution of prediction errors across 30 runs for DKASC and solar I datasets.**

The in-depth analysis of the prediction error distribution in Fig 13 provides strong evidence for the rationality and reliability of the model in uncertainty quantification. Each subplot in the figure includes a prediction error histogram, a non-parametric kernel density estimation (KDE) curve, and a Gaussian fitting curve.

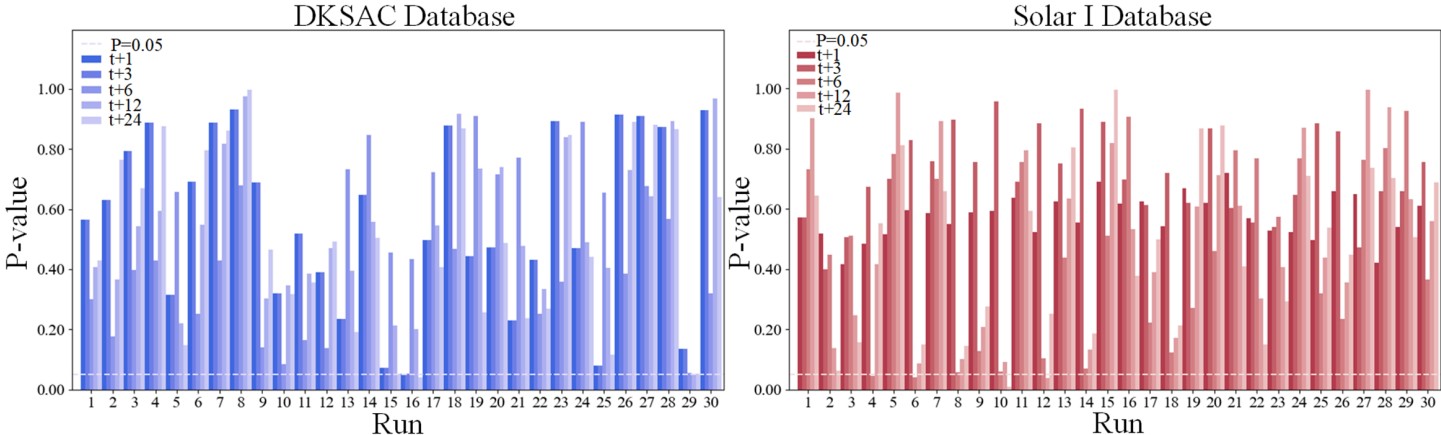

**Fig 11. Distribution of p-values across forecasting horizons for DKASC and solar I datasets.**

**Table 11. Prediction Interval Performance (PICP and PIAW) on DKASC and solar I datasets at different confidence levels.**

| Dataset | Confidence Level, % | PICP, % | | | | PIAW, kW | | | |
|---|---|---|---|---|---|---|---|---|---|
| | | 3 | 6 | 12 | 24 | 3 | 6 | 12 | 24 |
| DKASC | 95 | 96.56 | 95.89 | 95.70 | 94.01 | 0.5747 | 0.5806 | 0.6080 | 0.7678 |
| | 90 | 94.74 | 93.61 | 93.51 | 92.57 | 0.4666 | 0.4599 | 0.4825 | 0.5986 |
| | 85 | 92.86 | 91.59 | 92.01 | 91.11 | 0.3996 | 0.3991 | 0.4197 | 0.5165 |
| | 80 | 91.24 | 89.75 | 90.28 | 89.90 | 0.3475 | 0.3496 | 0.3709 | 0.4577 |
| | 75 | 89.29 | 87.79 | 88.15 | 88.23 | 0.3066 | 0.3092 | 0.3322 | 0.4053 |
| | 70 | 87.65 | 86.52 | 87.17 | 87.06 | 0.2731 | 0.2738 | 0.2995 | 0.3660 |
| | 65 | 85.68 | 84.85 | 84.87 | 83.72 | 0.2419 | 0.2472 | 0.2686 | 0.3260 |
| | 60 | 83.51 | 83.39 | 83.37 | 82.28 | 0.2163 | 0.2236 | 0.2407 | 0.2978 |
| Solar I | 95 | 96.85 | 94.91 | 94.16 | 90.44 | 5.5110 | 4.0990 | 3.9182 | 4.0399 |
| | 90 | 95.02 | 92.10 | 91.41 | 87.92 | 4.3886 | 3.2493 | 3.1660 | 3.2237 |
| | 85 | 93.25 | 89.98 | 89.81 | 85.46 | 3.7620 | 2.7845 | 2.7211 | 2.7060 |
| | 80 | 91.30 | 89.12 | 88.09 | 84.03 | 3.2735 | 2.4310 | 2.3899 | 2.3433 |
| | 75 | 87.92 | 87.24 | 85.86 | 80.19 | 2.8970 | 2.2000 | 2.1508 | 2.1086 |
| | 70 | 86.66 | 85.92 | 84.32 | 78.65 | 2.5802 | 1.9447 | 1.8944 | 1.8556 |
| | 65 | 82.83 | 83.40 | 82.08 | 74.59 | 2.3149 | 1.7531 | 1.7144 | 1.6734 |
| | 60 | 81.51 | 80.48 | 79.97 | 72.41 | 2.0733 | 1.5609 | 1.5057 | 1.4729 |

The analysis reveals two key points: firstly, all error distributions are centered at zero, indicating low systematic bias in the model predictions. Secondly, and more importantly, the KDE curves show that, especially in the two realistic and challenging scenarios of weather fluctuations and abrupt weather changes, the prediction errors exhibit significant non-Gaussian characteristics such as spikiness and heavy tails. By adopting quantile regression, the model is not constrained by traditional Gaussian assumptions but successfully captures and models this complex real-world error distribution. Additionally, the trend of the error distribution widening reasonably with the extension of the prediction horizon and the deterioration of weather conditions is highly consistent with the variation pattern of the PIAW metric, further confirming that the model proposed in this study can accurately and reliably capture the dynamic growth of uncertainties.

## 6 Conclusion

This study proposed a novel hybrid prediction model (ICIAL) that combines signal decomposition with a frequency-stratified network to improve photovoltaic (PV) power forecasting. The key findings of the research are as follows:

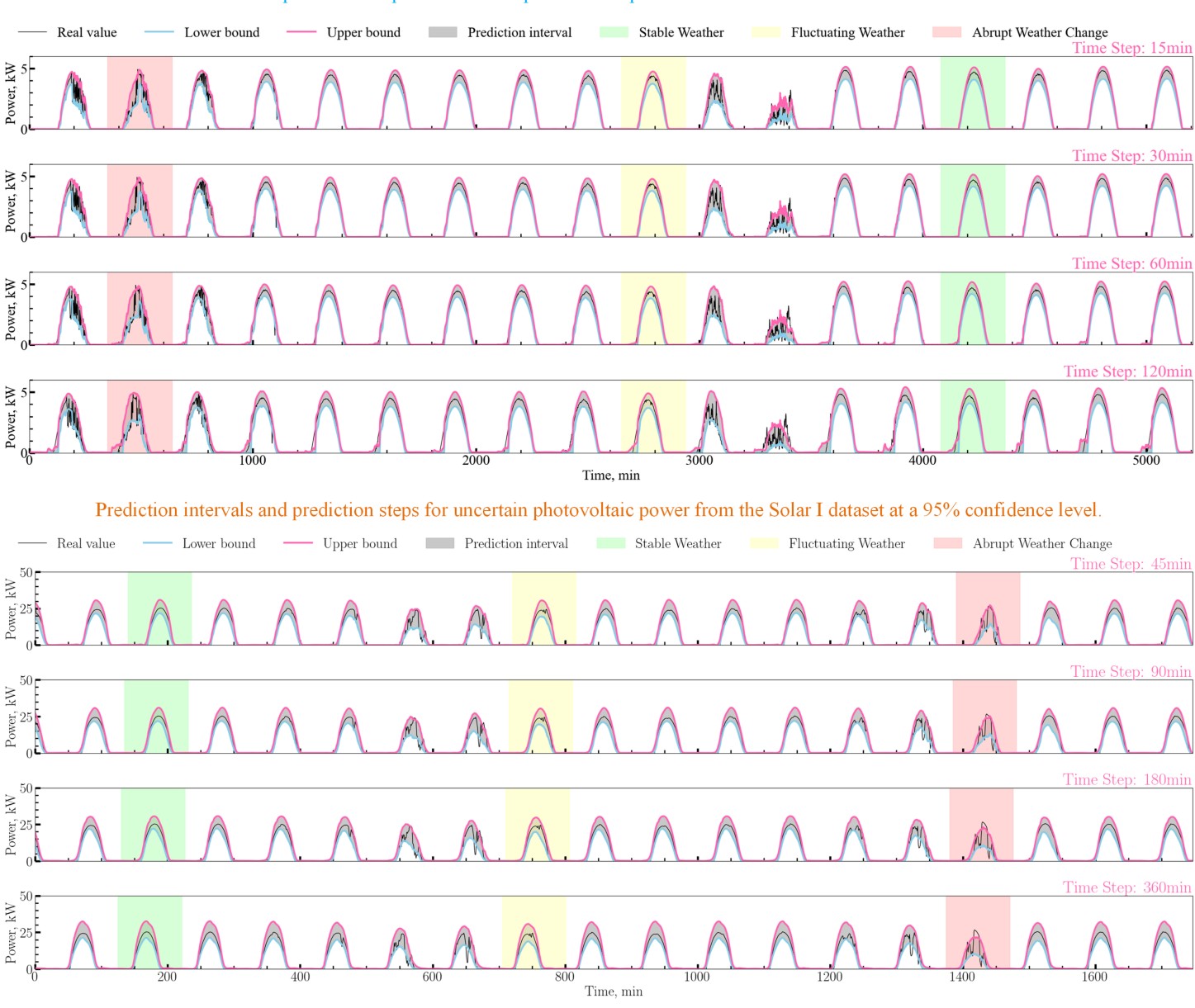

**Fig 12**. **Prediction intervals and prediction steps for uncertain photovoltaic power from the DKASC and solar I datasets at a 95% confidence level.**

1. The model first employs the ICEEMDAN algorithm to decompose the original photovoltaic power series into multi-scale components, effectively capturing the intrinsic characteristics of the data. Ablation experiments further confirm the efficacy of this method in improving prediction accuracy. Subsequently, a heterogeneous hierarchical network is implemented: an LSTM model is utilized to predict the low-frequency trend series, while a ConvBiGRU-A model, integrated with IRPE, is employed for the high-frequency error series. The integrated IRPE mechanism allows the model to adapt to external factors, such as changing light intensity, thereby maintaining high accuracy across various weather conditions.

**Prediction Error Distributions for Photovoltaic Power from the DKASC Dataset under Various Weather Conditions and at Different Prediction Horizons**

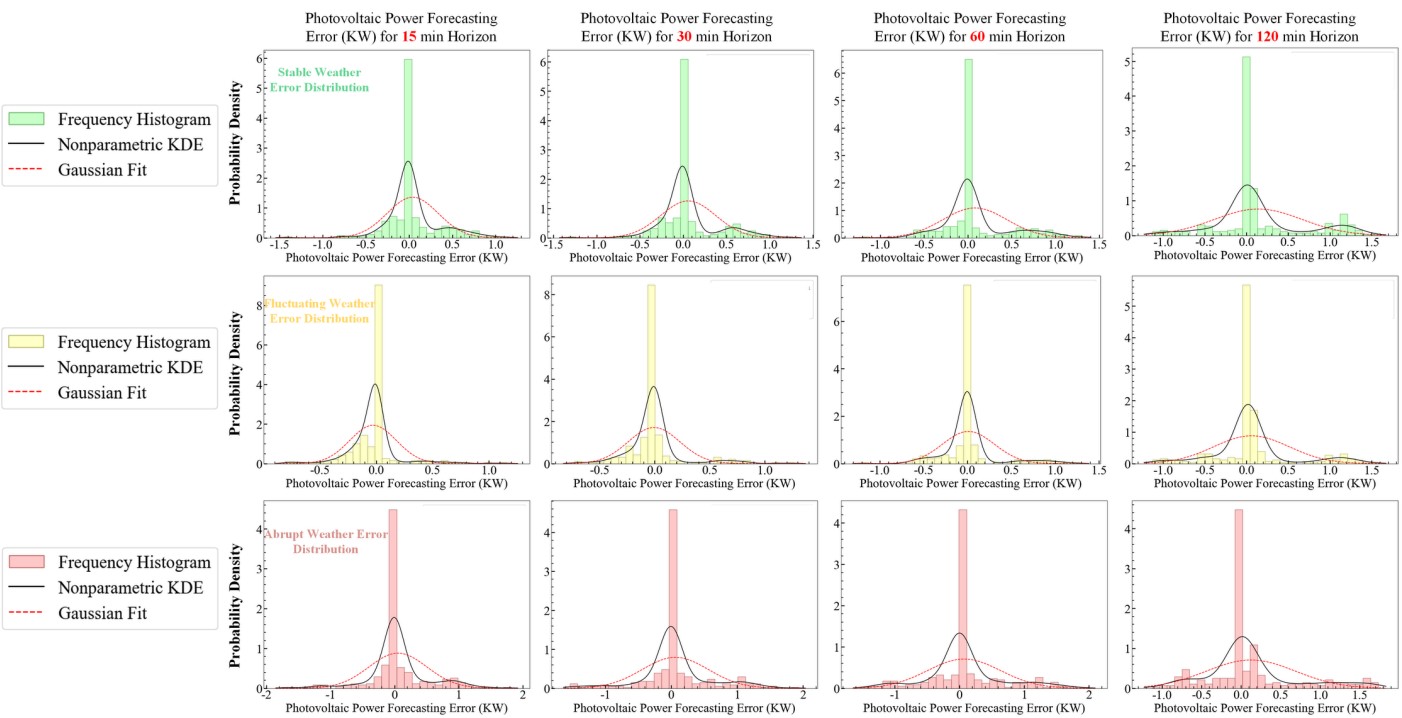

**Prediction Error Distributions for Photovoltaic Power from the Solar I dataset under Various Weather Conditions and at Different Prediction Horizons.**

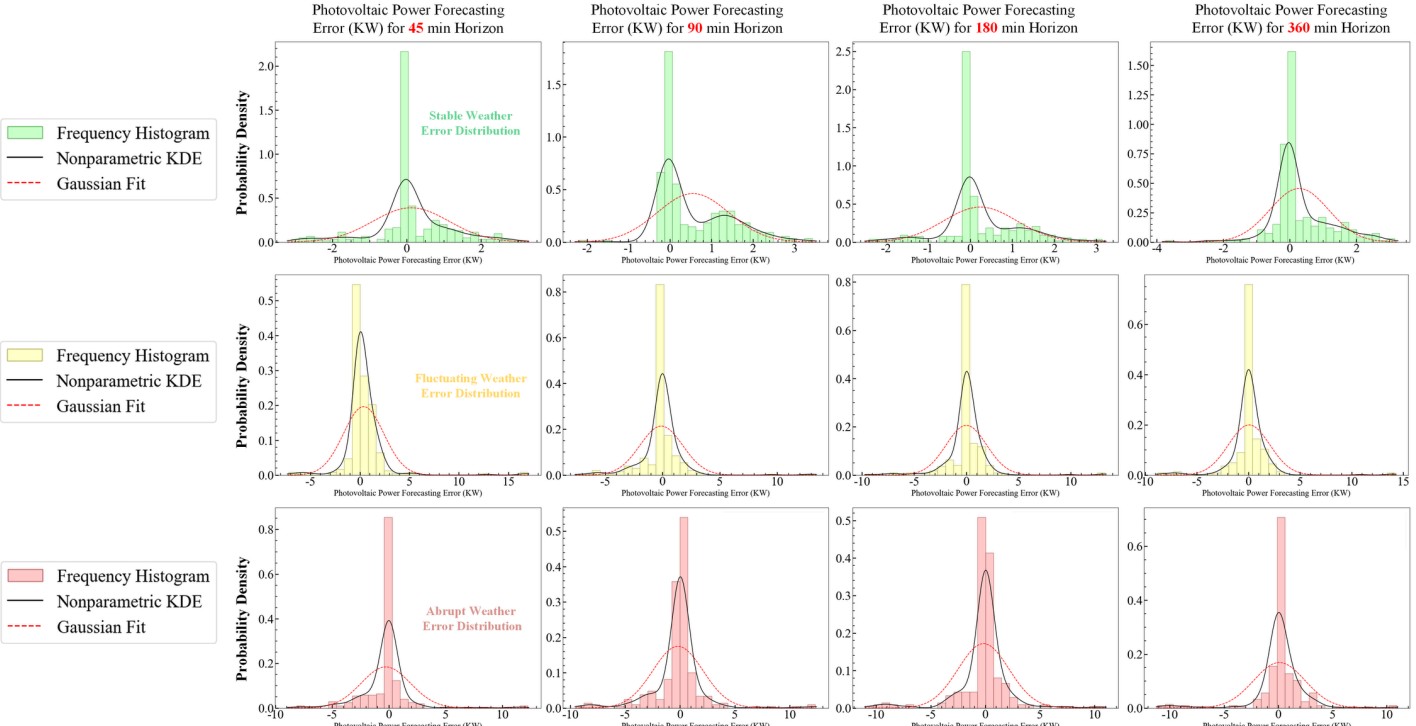

**Fig 13**. Prediction error distributions for photovoltaic power from the DKASC and solar I datasets under various weather conditions and at different prediction horizons.

2. By applying distinct models to high- and low-frequency components, this approach effectively captures complex PV dynamics, overcoming a key limitation of traditional methods. The method first employs the ICEEMDAN algorithm for signal decomposition and subsequently performs predictions through a hierarchical network that integrates IRPE mechanism. A visual analysis of the model's attention mechanism clearly reveals the effectiveness of IRPE: in critical scenarios such as peaks, valleys, and transient changes, the attention heatmaps of the ICIAL model exhibit sharper and more focused patterns, thereby verifying its ability to accurately identify and capture these complex dynamic changes. This advantage is ultimately reflected in the model's prediction performance. Comparative experimental results under different weather conditions show that, whether under stable, fluctuating, or drastically changing light conditions, the prediction curve of the ICIAL model closely aligns with the actual power values, particularly excelling in peak prediction accuracy and effectively addressing the common issue of peak underestimation observed in other baseline models.

3. Through comprehensive experiments conducted on different datasets (DKASC and Solar I) using three multi-step prediction strategies (MIMO, Direct, and Recursive), this study systematically verified the robustness of the proposed model. The results consistently indicated that the ICIAL model, under the MIMO strategy, exhibits strong long-term prediction capabilities. In the 120-minute and 2880-minute prediction tasks on the DKASC and Solar I datasets, its nMAE was reduced by 14.6% and 8.1%, respectively, compared to the second-best model. Furthermore, thirty independent rounds of the Wilcoxon signed-rank test confirmed the model's statistical stability, demonstrating consistent performance across multiple iterations.

4. To overcome the shortcomings of deterministic point predictions in quantifying future risks, this study introduces an uncertainty analysis framework based on quantile regression. Experiments have demonstrated that the model can generate reliable and sharp probabilistic prediction intervals. Whether applied to the DKASC dataset, which exhibits relatively stable data patterns, or the Solar I dataset, characterized by more complex weather conditions, the prediction intervals produced by the model strike a balance between a high PICP and a reasonable PIAW at different confidence levels. A comprehensive analysis of the prediction error distribution further indicates that the model effectively captures the non-Gaussian error characteristics commonly observed in the real world, confirming its practical value in power grid risk assessment and operational planning.

In conclusion, this study has fully demonstrated the outstanding performance of the ICIAL model in multi-step photovoltaic power prediction. By integrating the ICEEMDAN algorithm with a hierarchical frequency modeling neural network, the model has exhibited strong medium- to long-term prediction capabilities and robustness across diverse experimental scenarios. However, long-term prediction of photovoltaic power continues to pose a significant challenge for the efficient operation of power stations. Future work will focus on optimizing the model and extending its application to long-term forecasting scenarios.

Furthermore, the ICIAL model demonstrates broad generalization potential, with its core advantage stemming from its universal architectural principles rather than being confined to a specific domain. First, as a domain-agnostic decomposition tool, ICEEMDAN can effectively handle any non-stationary time series. Second, hierarchical frequency modeling—specifically, the customization of different models for low-frequency trends and high-frequency fluctuations—serves as a fundamental principle that is universally applicable to the analysis of complex systems.

Therefore, the framework could likely be transferred to other challenging prediction tasks. For instance, in the field of renewable energy, the model can be directly applied to wind power forecasting, which also faces the challenge of intermittency. Its potential applications can be extended to various domains that require high-precision predictions, such as grid load forecasting, financial market analysis (e.g., stock price forecasting), and climate modeling. The exploration of these applications will be a key direction for future research, aiming to further validate and expand the model's generalization capabilities and to contribute a robust analytical paradigm to the field of complex time series analysis.

## Nomenclature

| | |
|---|---|
| PV | Photovoltaic |
| ConvBiGRU | One-dimensional Convolutional Neural Network - Bidirectional Gated Recurrent Unit |
| IRPE | Improved Relative Position Encoding |
| LSTM | Long Short-Term Memory |
| ICEEMDAN | Improved Complete Ensemble Empirical Mode Decomposition with Adaptive Noise |
| ICIAL | ICEEMDAN-ConvBiGRU-IRPE-Attention-LSTM |
| SARIMA | Seasonal Autoregressive Integrated Moving Average |
| RVFL | Random Vector Function Link |
| MODWT | Maximum Overlap Discrete Wavelet Transform |
| LR | Linear Regression |
| RNN | Recurrent Neural Networks |
| PCC | Pearson Correlation Coefficient |
| PSO | Particle Swarm Parameter Optimisation |
| IMFs | Intrinsic Modal Functions |
| DHI | Diffuse Horizontal Irradiance |
| GHI | Global Horizontal Irradiance |
| EMD | Empirical Modal Decomposition |
| EEMD | Ensemble Empirical Modal Decomposition |
| CEEMDAN | Complete Ensemble Empirical Mode Decomposition |
| VMD | Variational Mode Decomposition |
| SVM | Support Vector Machines |
| BP | Back-Propagation |

## Author contributions

**Conceptualization:** Yaopeng Han, Jinghao Zhao.

**Data curation:** Yaopeng Han, Chenxi Li, Siqi Chen.

**Formal analysis:** Jinghao Zhao.

**Funding acquisition:** Yajun Tian.

**Investigation:** Yaopeng Han.

**Methodology:** Yaopeng Han, Jinghao Zhao.

**Project administration:** Jun Wang.

**Resources:** Chenxi Li, Siqi Chen.

**Supervision:** Jinghao Zhao, Yajun Tian.

**Validation:** Jinghao Zhao.

**Visualization:** Yaopeng Han.

**Writing – original draft:** Yaopeng Han.

**Writing – review & editing:** Jinghao Zhao, Yajun Tian, Jun Wang.

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
