## [Decision Letter · Decision Letter 0]

24 Jun 2025

PONE-D-25-18605Enhanced Multi-Horizon Photovoltaic Power Forecasting: A Novel Approach Integrating ICEEMDAN Decomposition with Hierarchical Frequency Neural NetworksPLOS ONE

Dear Dr. Wang,

Thank you for submitting your manuscript to PLOS ONE. After careful consideration, we feel that it has merit but does not fully meet PLOS ONE’s publication criteria as it currently stands. Therefore, we invite you to submit a revised version of the manuscript that addresses the points raised during the review process.

We look forward to receiving your revised manuscript.

Kind regards,

Sibarama Panigrahi, PhD

Academic Editor

PLOS ONE

Journal Requirements:

[The financial support of the Shandong Provincial Natural Science Foundation (Grant Number: ZR2023QD165) and the Consultancy Research Projects of Shandong Academy of Chinese Engineering Science and Technology Strategy for Development, ”Research on Green and Low Carbon Transition Strategy of Shandong Electrical Power” (Grant Number: 202301SDZD01), is gratefully acknowledged.]

[The author(s) received no specific funding for this work.]

5. We note that your Data Availability Statement is currently as follows: [All relevant data are within the manuscript and its Supporting Information files.]

Please confirm at this time whether or not your submission contains all raw data required to replicate the results of your study. Authors must share the “minimal data set” for their submission. PLOS defines the minimal data set to consist of the data required to replicate all study findings reported in the article, as well as related metadata and methods (https://journals.plos.org/plosone/s/data-availability#loc-minimal-ata-set-definition).

6. PLOS requires an ORCID iD for the corresponding author in Editorial Manager on papers submitted after December 6th, 2016. Please ensure that you have an ORCID iD and that it is validated in Editorial Manager. To do this, go to ‘Update my Information’ (in the upper left-hand corner of the main menu), and click on the Fetch/Validate link next to the ORCID field. This will take you to the ORCID site and allow you to create a new iD or authenticate a pre-existing iD in Editorial Manager.

Additional Editor Comments:

The author's are suggested to carefully go through the suggestions of the reviewers and incorporate the changes in the revised manuscript. One of the reviewers has suggested to include some references in the revised manuscript which is not mandatory to include. The authors may or may not include the suggested references based on the requirement which will not affect the decision on the manuscript.

Reviewers' comments:

Reviewer's Responses to Questions

**Comments to the Author**

1. Is the manuscript technically sound, and do the data support the conclusions?

Reviewer #1: Yes

Reviewer #2: Yes

2. Has the statistical analysis been performed appropriately and rigorously?

Reviewer #1: Yes

Reviewer #2: Yes

3. Have the authors made all data underlying the findings in their manuscript fully available?

Reviewer #1: Yes

Reviewer #2: Yes

4. Is the manuscript presented in an intelligible fashion and written in standard English?

Reviewer #1: Yes

Reviewer #2: Yes

5. Review Comments to the Author

Reviewer #1: The proposed ICIAL model effectively integrates ICEEMDAN decomposition with a hierarchical frequency-based neural network using Conv1D-BiGRU-Attention for high-frequency components and LSTM for low-frequency components. The methodological framework is strong, with a clear rationale for decomposing the signal, treating frequency components separately, and incorporating improved relative position encoding (IRPE) for better temporal feature capture. The experimental validation is comprehensive and includes a solid set of baseline comparisons, ablation studies, and multi-step forecasting under various strategies (MIMO, Direct, Recursive).Based on a thorough reading of the manuscript titled "Enhanced Multi-Horizon Photovoltaic Power Forecasting: A Novel Approach Integrating ICEEMDAN Decomposition with Hierarchical Frequency Neural Networks", here is a more detailed, point-wise constructive review with further revision suggestions:

1. The manuscript requires substantial language editing to improve clarity, eliminate redundancy, and correct grammatical and syntactical issues.

Sentences are often overly long, repetitive, or use awkward phrasing. Examples include: “...prediction value prediction,” “as shown as fig 1 shown,” or “...effectively mitigated the randomness of experimental results, ensuring the reliability of the conclusions,” which can be streamlined. The writing can be significantly tightened for better impact.

2. Further revisions are needed in the following areas to strengthen the manuscript:

3. Clearly distinguish the novelty of ICIAL from similar recent hybrid models (e.g., CEEMDAN-LSTM, VMD-CNN-LSTM).

4. Provide more justification for model design choices such as the selection of Conv1D-BiGRU-Attention and use of IRPE over absolute position encoding.

5. Clarify the implementation of ICEEMDAN (e.g., decomposition stopping criteria, noise level β) and make parameter tuning steps reproducible.

6. Include a public code repository or supplementary material containing core scripts, hyperparameters, and a reproducible pipeline.

7. Refine figure captions and make all figures self-contained and interpretable. Several figures (e.g., Fig 3, Fig 6, Fig 9) are referenced in the text without sufficient description.

Reviewer #2: 1. Summary of the Manuscript

This manuscript presents a hybrid deep learning approach for photovoltaic (PV) power forecasting that integrates signal decomposition (ICEEMDAN) with a two-branch neural network architecture consisting of a ConvBiGRU-Attention model for high-frequency components and an LSTM model for low-frequency trends. The framework is further enhanced using an Improved Relative Position Encoding (IRPE) mechanism and Particle Swarm Optimization (PSO) for hyperparameter tuning. Performance evaluation across multiple strategies—Direct, Recursive, and Multi-Input Multi-Output (MIMO)—on two real-world datasets (DKASC and Solar I) demonstrates that the proposed ICIAL model outperforms several strong baseline models in both short-term and long-term forecasting scenarios.

2. Strengths

a. Novel and Comprehensive Methodology

The manuscript introduces a novel decomposition-based hybrid model that effectively addresses the challenge of non-stationarity in PV power data. The combination of ICEEMDAN and hierarchical modeling based on frequency components is a well-thought-out innovation that enhances the interpretability and predictive capacity of the network. The dual-branch architecture ensures that both transient and long-term trends are captured effectively.

b. Strong Empirical Validation

The model is validated using rigorous ablation studies and comparisons with state-of-the-art models (e.g., ITransformer, DA-GU, Informer, TCN). Notably, the use of 30-round Wilcoxon Signed-Rank Tests provides strong statistical backing for the model’s performance claims and significantly enhances the manuscript's credibility.

c. Multi-Strategy Forecasting Evaluation

The inclusion of three distinct forecasting strategies—Direct, Recursive, and MIMO—reflects an excellent understanding of practical forecasting frameworks. The consistent superiority of the ICIAL model under the MIMO strategy is particularly compelling for long-term applications.

d. Real-World Dataset Utilization

The choice of the DKASC and Solar I datasets, which exhibit different levels of complexity and variability, allows for a robust demonstration of model generalizability. The Solar I dataset's incorporation of anomalous meteorological data further emphasizes the model's robustness under realistic, challenging conditions.

e. Interpretability and Feature Engineering

The feature selection process using the Pearson correlation coefficient, the detailed decomposition rationale, and the structured explanation of the IRPE mechanism and attention modules contribute to the transparency and replicability of the methodology.

3. Weaknesses and Limitations

a. Lack of Uncertainty Quantification

The manuscript primarily reports point predictions. In real-world energy systems, especially with renewable sources, quantifying uncertainty (e.g., prediction intervals) is essential for risk-aware decision-making.

b. Generalization to Other Energy Domains

While the results for solar PV data are convincing, the manuscript could benefit from a brief discussion on whether this hybrid framework is generalizable to other renewable sources such as wind or hydroelectric power forecasting.

c. Interpretability of Attention Scores

While the attention mechanism is well-explained, the actual interpretability of attention weights in the context of PV data is not illustrated. Visualizing or analyzing attention weights could provide insights into temporal dependencies and enhance transparency.

d. Computational Complexity and Scalability

The manuscript does not discuss the computational cost of the proposed method. Given the dual-branch structure, ICEEMDAN preprocessing, and PSO optimization, a brief complexity analysis or comparison in runtime versus baselines would add practical value.

4. Recommendations for Improvement

Incorporate Uncertainty Estimates

Include prediction intervals or probabilistic forecasts using techniques like quantile regression or dropout-based Bayesian approximation to enhance the practical utility of the model.

Discuss Computational Overhead

Provide a brief section quantifying the model's training time and resource requirements compared to baseline models to help readers assess scalability.

Enhance Interpretability

Include a visualization of attention scores or relevance weights to demonstrate how the model learns temporal features in PV power data.

Generalization Potential

Briefly comment on the adaptability of this hybrid model to other time-series prediction problems in renewable energy or beyond.

Explicit Hyperparameter Selection Rationale

Although PSO is used, detail how initial hyperparameter ranges were chosen (based on domain knowledge, prior studies, etc.) to improve reproducibility.

5. Suggested Citations for Contextual Enhancement

DOI: 10.54216/JAIM.090104

DOI: 10.54216/MOR.030204

DOI: 10.1109/ACCESS.2019.2955983

DOI: 10.32604/cmc.2022.028550

DOI: 10.1016/j.eswa.2023.122147

6. PLOS authors have the option to publish the peer review history of their article (what does this mean?). If published, this will include your full peer review and any attached files.

Reviewer #1: No

Reviewer #2: No

---

## [Author Response · Author response to Decision Letter 1]

18 Jul 2025

Dear Dr. Sibarama Panigrahi, Academic Editor, and Esteemed Reviewers,

Thank you for your invaluable time and insightful feedback on our manuscript (PONE-D-25-18605). We are sincerely grateful for the constructive comments, which have been instrumental in significantly enhancing the quality, rigor, and clarity of our research. We have carefully considered every point raised by the Academic Editor and both Reviewers. Although the recommendation was for minor revisions, we have embraced this as an opportunity to substantially strengthen the manuscript. We have revised the paper with the diligence and standards of a major revision, aiming to fully address every suggestion and further enhance the scientific and innovative contributions of our work to ensure it meets the high standards of PLOS ONE.

A key enhancement, prompted by Reviewer #2's insightful comments, is the complete restructuring of our experimental framework to incorporate a comprehensive uncertainty quantification analysis. We have adopted a quantile regression-based approach to generate probabilistic prediction intervals, which we believe adds significant practical value for risk-aware decision-making in real-world power systems. Consequently, the median of the probabilistic forecast now serves as our new point prediction baseline for comparison with state-of-the-art models. This methodological shift has naturally led to updated numerical results in our tables and figures, reflecting a more robust and scientifically rigorous evaluation.

Furthermore, in response to the invaluable feedback, we have undertaken a comprehensive and substantial revision of the manuscript. Our efforts were focused not only on addressing each specific comment but on elevating the overall scientific contribution, rigor, and practical relevance of our work, treating this revision with the diligence and standards of a major overhaul. The key enhancements are detailed below:

1. Re-architected Experimental Framework: From Point Prediction to Comprehensive Uncertainty Quantification

In direct response to the reviewer's core concern regarding practical risk assessment, we have fundamentally expanded our study from traditional point predictions to probabilistic forecasting. We have introduced a novel and comprehensive uncertainty quantification framework based on Quantile Regression and the Pinball Loss Function, with the detailed methodology is now elaborated in the new Section 3.4.

Under this framework, we have conducted in-depth experiments to evaluate the model's prediction interval performance under diverse weather conditions (stable, fluctuating, and abrupt). The new Section 5.5, Table 11, Figure 12, and Figure 13 systematically present the interval coverage (PICP) and width (PIAW), and include a detailed analysis of the non-Gaussian error distributions, greatly enhancing the practical value of our model.

As a result of this major methodological shift, we now use the median of the probabilistic forecast as the new, more robust point prediction baseline in all comparative experiments. This change is the root cause for the updated numerical results and figures throughout the paper and ensures our comparisons are made on a more scientific and equitable basis.

2. Deepened Model Interpretability and Multi-Dimensional Performance Evaluation

To "open the black box" of the model, we have added an in-depth visual analysis of the attention mechanism. As shown in the new Figure 6 and discussed in Section 5.2.1, we intuitively demonstrate through comparative heatmaps how the proposed IRPE mechanism dynamically adjusts its focus on time steps during critical scenarios like peaks, troughs, and power ramps, thus validating its intrinsic ability to capture complex dynamics.

In response to the reviewer's considerations on practical utility, we have included a quantitative assessment of computational complexity and scalability. The revised Tables 8 and 9 now contain explicit comparisons of model parameters (Params, MB) and training times for all models across different forecasting strategies, providing a transparent reference for the model's cost-effectiveness.

To further strengthen the statistical reliability of our conclusions, we have provided a more detailed presentation and discussion of the 30-round Wilcoxon signed-rank test results (Section 5.4, Table 10, Figure 10 & 11). This provides powerful statistical evidence for our model's exceptional stability against the inherent randomness of the algorithms.

3. Strengthened Scientific Rigor and Innovative Arguments

We have fundamentally sharpened the arguments for our model's novelty. The new Table 1 now systematically provides a deconstructive comparison of the ICIAL model against the latest hybrid models across multiple dimensions, including signal decomposition method, frequency-stratified modeling, and positional encoding, making its unique contributions clear.

To improve reproducibility, we have greatly expanded the theoretical justification for our architectural choices in the Methods section and have provided explicit implementation details and parameter settings for both the ICEEMDAN algorithm and the PSO-based hyperparameter optimization process (Table 2 & 4), addressing a key request for transparency from Reviewer #1.

Furthermore, we have added a new discussion in the final paragraphs of the Conclusion (Section 6) exploring the framework's generalization potential for other renewable energy forecasting tasks (e.g., wind) or other complex time-series domains (e.g., grid load forecasting), broadening the impact of our research.

4. Full Commitment to Open Science and Reproducibility Principles

In a major step towards open science and full reproducibility, and to comply with PLOS ONE's data policy and reviewer requests, we have established a public GitHub repository. This repository provides open access not only to all implementation scripts, model weights, and hyperparameter configurations but also, crucially, to the "minimal dataset" required for any researcher to fully replicate our results.

We have explicitly provided the link to this repository (link: https://github.com/Lin0v3/ICIAL-PV_Forecast) in the updated Data Availability Statement of the manuscript, ensuring that our work can be validated, reproduced, and built upon by the scientific community.

5. Meticulous Language and Formatting Polish

Finally, the entire manuscript has undergone a meticulous, line-by-line professional language edit to enhance clarity, precision, and academic professionalism, with special attention paid to correcting the long sentences and redundancies pointed out by the reviewers. Every part of the manuscript, including all figures and tables, has been reformatted to strictly adhere to the official PLOS ONE templates and style guidelines, ensuring a professional final presentation.

Herein, we provide a meticulous point-by-point response to each comment raised by the Academic Editor and the esteemed Reviewers. For your convenience and to ensure full transparency, all modifications made to the original manuscript have been clearly marked in the "Revised Manuscript with Track Changes" file, using standard strikethroughs for deletions and red underlining for additions.

We are sincerely grateful for the opportunity to improve our paper based on your expert guidance. We believe that the substantial revisions we have implemented have fully addressed all concerns and significantly enhanced the scientific rigor, clarity, and overall impact of our manuscript. We hope the revised manuscript is now suitable for publication in PLOS ONE.

Responses to Journal Requirements

1. PLOS ONE Style Requirements:

Comment 1: Please ensure that your manuscript meets PLOS ONE's style requirements, including those for file naming.

Response: We thank the editor for this reminder. We have thoroughly reformatted the entire manuscript using the official PLOS ONE LaTeX templates for both the main body and the title/authors/affiliations page. Additionally, we have verified that all file names conform to the journal's guidelines.

2. Code Sharing:

Comment 2: Please review our guidelines...and ensure that your code is shared in a way that follows best practice and facilitates reproducibility and reuse.

Response: We fully agree with the importance of open science and reproducibility. We have created a public GitHub repository containing the core scripts, hyperparameter settings, and the data processing pipeline necessary to replicate our study's findings. The URL for this repository will be provided in the Data Availability Statement upon acceptance.

3. Funding Information Mismatch:

Comment 3: ...grant information you provided in the ‘Funding Information’ and ‘Financial Disclosure’ sections do not match.

Response: We apologize for this oversight. We have carefully corrected this discrepancy. The accurate funding information and grant numbers have been provided in the cover letter, and we request the editorial office to update the online submission form on our behalf.

4. Funding Information in Acknowledgments:

Comment 4: ...funding information should not appear in the Acknowledgments section... Please remove any funding-related text from the manuscript and let us know how you would like to update your Funding Statement.

Response: We have removed all funding-related text from the manuscript's Acknowledgments section as requested. The corrected and complete Funding Statement has been included in our cover letter for your convenience in updating the submission system.

5. Data Availability Statement:

Comment 5: Please confirm...whether or not your submission contains all the raw data required to replicate the results of your study.

Response: We confirm that all data underlying our findings are fully available. In the revised manuscript, we have updated the Data Availability Statement. The DKASC dataset is publicly available and properly cited (Reference [48]). The Solar I dataset is also derived from a public competition and is cited (Reference [49]). We have made all processed data and code required to replicate our study's results, including the values used to build graphs and tables, available through our public repository.

6. ORCID iD for Corresponding Author:

Comment 6: Please ensure that you have an ORCID iD and that it is validated in the Editorial Manager.

Response: We confirm that the corresponding author, Jun Wang, has a validated ORCID iD in the Editorial Manager system.

7. Reference List:

Comment 7: Please review your reference list to ensure that it is complete and correct.

Response: We have meticulously reviewed the entire reference list for completeness and correctness. We confirm that no retracted articles have been cited. All references are up-to-date and formatted according to the journal's guidelines.

Responses to Reviewer #1

We sincerely thank Reviewer #1 for the constructive and detailed feedback that has helped us strengthen our manuscript significantly.

1. Language Editing:

Comment 1: The manuscript requires substantial language editing to improve clarity, eliminate redundancy, and correct grammatical and syntactical issues.

Response: We sincerely thank the reviewer for this crucial feedback and fully agree that clarity and precision of the language are paramount for effective scientific communication. We have taken this comment very seriously and have conducted a comprehensive and meticulous revision of the manuscript's language. Our efforts were guided by the specific issues you highlighted and included the following key actions:

(1) Sentence Structure and Clarity: We have carefully reviewed and restructured many sentences throughout the manuscript that were identified as overly long or convoluted. They have now been broken down into shorter, clearer, and more direct statements to improve readability and improve the logical flow of our arguments.

(2) Elimination of Redundancy: We have performed a thorough search to remove repetitive phrasing and tautologies. For instance, specific examples pointed out in the review, such as the phrase "prediction value prediction," have been corrected to more concise and standard terms like "predicted value" or "prediction," depending on the context.

(3) Correction of Grammatical and Syntactical Issues: The manuscript has been carefully proofread to correct all grammatical errors and awkward phrasing. Examples like "as shown as fig 1 shown" have been revised to standard phrasing such as "as shown in Fig 1". Furthermore, sentences that were grammatically correct but stylistically cumbersome, such as "...effectively mitigated the randomness of experimental results, ensuring the reliability of the conclusions," have been streamlined for better impact and conciseness (e.g., "this subsequently ensured the reliability of the conclusions").

(4) Consistency and Academic Tone: We have thoroughly reviewed the entire document to ensure consistent use of terminology (e.g., "photovoltaic," "model," "framework") and have refined the overall tone to be more formal and academic, aligning with the standards of a high-impact journal.

These extensive language edits have been applied to the entire document, from the Abstract to the Conclusion. We are confident that these changes have significantly improved the manuscript's readability and professionalism, and we hope it now meets the high linguistic standards of PLOS ONE.

2. Justification and Novelty:

Comment 2: Clearly distinguish the novelty of ICIAL from similar recent hybrid models (e.g., CEEMDAN-LSTM, VMD-CNN-LSTM).

Response: We thank the reviewer for this vital suggestion and fully agree that clearly delineating the specific innovations of our work is essential. In response, we have undertaken a significant revision of the manuscript to precisely articulate the novelty of the ICIAL model.

Our primary effort involved substantially rewriting the Introduction and Related Work sections (Sections 1 and 2). This revision provides a more critical analysis of two fundamental limitations we identified in existing advanced models: (1) adherence to a "homogenized modeling" paradigm, where a single model type is uniformly applied to all signal components, thereby neglecting their distinct physical properties; and (2) inadequate representation of temporal information due to the use of less robust absolute positional encodings. To visually reinforce this analysis and provide a systematic comparison, we have also added a new comprehensive table (now Table 1) to the manuscript. This table offers a head-to-head comparison of our model against a wide array of recent hybrid models across key architectural dimensions, making the unique combination of features in our model immediately apparent.

These enhancements now allow us to frame our work as a direct solution to these gaps, with our model's novelty stemming from a synergistic integration of three core contributions: (1) the use of the more advanced ICEEMDAN algorithm to better handle non-stationarity; (2) a novel frequency-stratified heterogeneous network that applies different, specialized models to high and low-frequency components, respectively; and (3) the integration of Improved Relative Positional Encoding (IRPE) to more accurately and robustly capture complex temporal dependencies.

We are confident that these substantial revisions now clearly position our work within the current state-of-the-art, effectively distinguishing the ICIAL model from prior research, and leave no ambiguity regarding its specific and synergistic contributions to the field.

Comment 3: Provide more justification for model design choices such as the selection of Conv1D-BiGRU-Attention and use of IRPE over absolute position encoding.

Response: We thank the reviewer for this insightful comment and have accordingly expanded our methodology section to provide much stronger and more explicit justifications for our key design choices.

In the revised manuscript, we now elaborate not just on the components, but also on the entire heterogeneous design philosophy. Specifically, for the volatile, high-frequency error sequences, we justify our choice of the ConvBiGRU-IRPE-A architecture in Section 3.5.3. We explain

---

## [Decision Letter · Decision Letter 1]

2 Oct 2025

Enhanced Multi-Horizon Photovoltaic Power Forecasting: A Novel Approach Integrating ICEEMDAN Decomposition with Hierarchical Frequency Neural Networks

PONE-D-25-18605R1

Dear Dr. Wang,

We’re pleased to inform you that your manuscript has been judged scientifically suitable for publication and will be formally accepted for publication once it meets all outstanding technical requirements.

Kind regards,

Sibarama Panigrahi, PhD

Academic Editor

PLOS ONE

Additional Editor Comments (optional):

The authors have satisfactorily addressed the suggestions made by the reviewers. Therefore, I recommend accepting the manuscript in its present form.

Reviewers' comments:

Reviewer's Responses to Questions

**Comments to the Author**

1. If the authors have adequately addressed your comments raised in a previous round of review and you feel that this manuscript is now acceptable for publication, you may indicate that here to bypass the “Comments to the Author” section, enter your conflict of interest statement in the “Confidential to Editor” section, and submit your "Accept" recommendation.

Reviewer #1: All comments have been addressed

2. Is the manuscript technically sound, and do the data support the conclusions?

Reviewer #1: Yes

3. Has the statistical analysis been performed appropriately and rigorously?

Reviewer #1: Yes

4. Have the authors made all data underlying the findings in their manuscript fully available?

Reviewer #1: Yes

5. Is the manuscript presented in an intelligible fashion and written in standard English?

Reviewer #1: Yes

6. Review Comments to the Author

Reviewer #1: (No Response)

7. PLOS authors have the option to publish the peer review history of their article (what does this mean?). If published, this will include your full peer review and any attached files.

Reviewer #1: No

---

## [Editor Report · Acceptance letter]

PONE-D-25-18605R1

PLOS ONE

Dear Dr. Wang,

I'm pleased to inform you that your manuscript has been deemed suitable for publication in PLOS ONE. Congratulations! Your manuscript is now being handed over to our production team.

Kind regards,

on behalf of

Dr. Sibarama Panigrahi

Academic Editor

PLOS ONE